# Massively parallel reporter perturbation assays uncover temporal regulatory architecture during neural differentiation

Anat Kreimer[1,2,3,4,8]✉, Tal Ashuach[3,8], Fumitaka Inoue [1,2,5,8], Alex Khodaverdian[3], Chengyu Deng[1,2], Nir Yosef [3,6,7]✉ & Nadav Ahituv [1,2]✉

Gene regulatory elements play a key role in orchestrating gene expression during cellular differentiation, but what determines their function over time remains largely unknown. Here, we perform perturbation-based massively parallel reporter assays at seven early time points of neural differentiation to systematically characterize how regulatory elements and motifs within them guide cellular differentiation. By perturbing over 2,000 putative DNA binding motifs in active regulatory regions, we delineate four categories of functional elements, and observe that activity direction is mostly determined by the sequence itself, while the magnitude of effect depends on the cellular environment. We also find that fine-tuning transcription rates is often achieved by a combined activity of adjacent activating and repressing elements. Our work provides a blueprint for the sequence components needed to induce different transcriptional patterns in general and specifically during neural differentiation.

[1] Department of Bioengineering and Therapeutic Sciences, University of California, San Francisco, San Francisco, CA 94158, USA. [2] Institute for Human Genetics, University of California, San Francisco, San Francisco, CA 94158, USA. [3] Department of Electrical Engineering and Computer Sciences and Center for Computational Biology, University of California, Berkeley, CA 94720, USA. [4] Department of Biochemistry and Molecular Biology, Center for Advanced Biotechnology and Medicine, Robert Wood Johnson Medical School, Rutgers University, Piscataway, NJ 08854, USA. [5] Institute for the Advanced Study of Human Biology (WPI-ASHBi), Kyoto University, Kyoto 606-8501, Japan. [6] Chan-Zuckerberg Biohub, San Francisco, CA 94158, USA. [7] Ragon Institute of MGH, MIT, and Harvard, Cambridge, MA 02139, USA. [8] These authors contributed equally: Anat Kreimer, Tal Ashuach, Fumitaka Inoue. ✉email: kreimer@cabm.rutgers.edu; niryosef@berkeley.edu; nadav.ahituv@ucsf.edu

Enhancers are DNA sequences containing clustered recognition sites (i.e., motifs) for transcription factors (TFs) that play a pivotal role in transcriptional regulation of gene expression during numerous biological processes, including cellular differentiation[1]. This is evident by the abundance of disease-associated variants discovered through genome-wide association studies (GWAS) and expression quantitative trait loci (eQTLs) residing in noncoding regions[2]. Despite their importance, our understanding of the regulatory grammar of enhancers, namely the manner by which their DNA sequences pertain to their function remains largely unknown, thus limiting our ability to infer how changes in these sequences affect their functionality and lead to higher-level consequences.

Various biochemical assays (e.g., ChIP-seq, DNase-seq, ATAC-seq) have enabled genome-wide identification and characterization of candidate regulatory sequences such as enhancers, across different cell types[3], providing descriptive maps of the human genome. Complementary studies use genome modification approaches, such as CRISPR-Cas9, to functionally characterize enhancer elements by targeting their locations in the genome[4]. Such assays capture both direct and indirect causal relationships between the tested regulatory elements and cellular phenotype (e.g., gene expression) and in many cases target regions that are bound by specific transcription factors of interest[5]. Massively parallel reporter assays (MPRAs) provide an alternative approach that enables simultaneously testing the regulatory activity of thousands of regulatory sequences and their variants. In MPRA, a sequence of interest is synthesized and placed in front of a transcribed barcode. There are many variants to this technology[6], including one that utilizes lentivirus to integrate into the genome (hereafter we refer to this as lentiMPRA;[7]) used for these assays. The ratio between the abundance of a transcribed barcode (read with RNA-seq) and the number of coding sequences (evaluated with DNA-sequencing) provides a quantitative readout for the regulatory activity of the assayed sequence[6,8–12].

Approaches to understanding the roles of TFs in determining the activity of a given enhancer and the interplay between TFs in an enhancer[13] are generally limited by the number of causal relationships they can study directly (e.g., via gene knockdown), primarily due to cost and availability of efficient perturbing agents. Therefore large-scale studies often use correlational inference, e.g., associating TF binding with changes in gene expression based on motif- gene association[14]. These, however, are confounded by a slew of observations whereby only a small fraction of potential TF-binding sites (TFBSs) are actually occupied in any given cell-type, and these sites vary substantially across cell types and conditions[15–18]. Another caveat of perturbing endogenous factors that affect gene expression (e.g., TFs, enhancer regions) is the abundance of indirect effects, which are difficult to discern from the direct ones. These two issues are mitigated by MPRAs, as it provides a cost-effective approach to investigate thousands of candidate enhancer sequences along with variants of these sequences in which certain DNA-binding motifs are perturbed. The concern for indirect effects is mitigated to some extent as well due to the synthetic nature of the assay (i.e., the transcribed barcode is non-functional). Previous approaches to perturbation MPRA for sequence motifs were limited to several factors and a specific cell-type or condition. For example, a previous study[9,19] explored the activity of five activator motifs and two repressor motifs in K562 and HepG2 cells by introducing different variations to the motif sequence and another study[19] disrupted a single motif (PPARγ) in mouse adipocytes. Altogether, they focused on a specific time point and not a temporal course or developmental process.

The differentiation of stem cells into a neural lineage provides an exemplary model for studying how gradual and non-reversible changes to the cell's phenotype may be transcriptionally regulated. During this process, stem cells rapidly differentiate both on a molecular and physiological level to generate neurons. We previously characterized the temporal dynamics of gene expression (RNA-seq) and gene regulation (ATAC-seq, H3K27ac and H3K27me3 ChIP-seq and lentiMPRA) at seven time points (0–72 h) during the early parts of this process[20]. Using lentiMPRA, we identified numerous endogenous sequences that had temporal enhancer activity (i.e., the expression of their target barcode was well over the background levels and significantly changed over time). This activity tended to correlate with cell-endogenous changes to the expression of their target gene and to the structure of their surrounding chromatin. In addition, the genomic positions of the validated temporal sequences significantly overlapped with loci that have been associated with neurodevelopmental disorders, in particular autism spectrum disorder (ASD). Combining all our genomic data, we developed a prioritization method to select TFs that are putatively involved in driving a neural fate, and validated the role of several candidates with direct genetic perturbations. This study, however, was still limited to validations of a handful of TFs and lacks in understanding of the way by which these TFs may drive changes in transcription over time.

To more comprehensively identify DNA-binding motifs that may affect transcription and characterize the timing in which they carry out their effect, we utilized a 'perturbation MPRA' approach. Based on our previous data, we compiled a list of 591 regulatory sequences whose activity differed over time (considering different temporal patterns) as well as a selected set of 255 motifs within those regions. We then prioritized for testing 2144 instances of the selected motifs in the selected regions. We used leniMPRA to perturb, via three different approaches, the selected instances, and evaluated their effect over at the same seven time points (0–72 h) during the neural differentiation process. Using this approach, we found that 27% (598) of the perturbations had a significant effect on the transcription of the reporter gene. We divided these motif instances into several subtypes based on the direction (suppressing or inducing transcription) and magnitude (fold change, compared to the unperturbed and negative control sequences) of their effect. We observed that the magnitude of the effects often varied over time (indicating that it depends on the cellular environment), while the direction of the effect is independent of time and is broadly determined by the DNA sequence (i.e the combination of the perturbed motif and the surrounding region). Furthermore, we observed cases of activating and repressing motif instances that are harbored within the same regulatory region, suggesting that in those cases, fine-tuning of transcription levels may be achieved by a combination of opposing effects. Finally, by perturbing pairs of motifs in a select set of sequences, we found evidence for different patterns of cooperation between motifs, and that both fundamental models, namely the "enhanceosome" model of an all-or-nothing machinery, and the "billboard" model of independent contribution[15,21] are supported by our data. Overall, our findings suggest that the regulatory grammar of enhancers that changed in gene expression in our system is an amalgam of a wide variety of different mechanisms. It also helps establish perturbation MPRA as a powerful approach for high-throughput investigation of such mechanisms in different cellular contexts.

## Results

**Selection of regions and motifs for perturbation MPRAs**. To characterize the effect of DNA-binding motifs on gene expression over time, we first set out to choose a set of regulatory regions that showed temporal activity during early neural differentiation using lentiMPRA data from our previous study (0, 3, 6, 12, 24, 48,

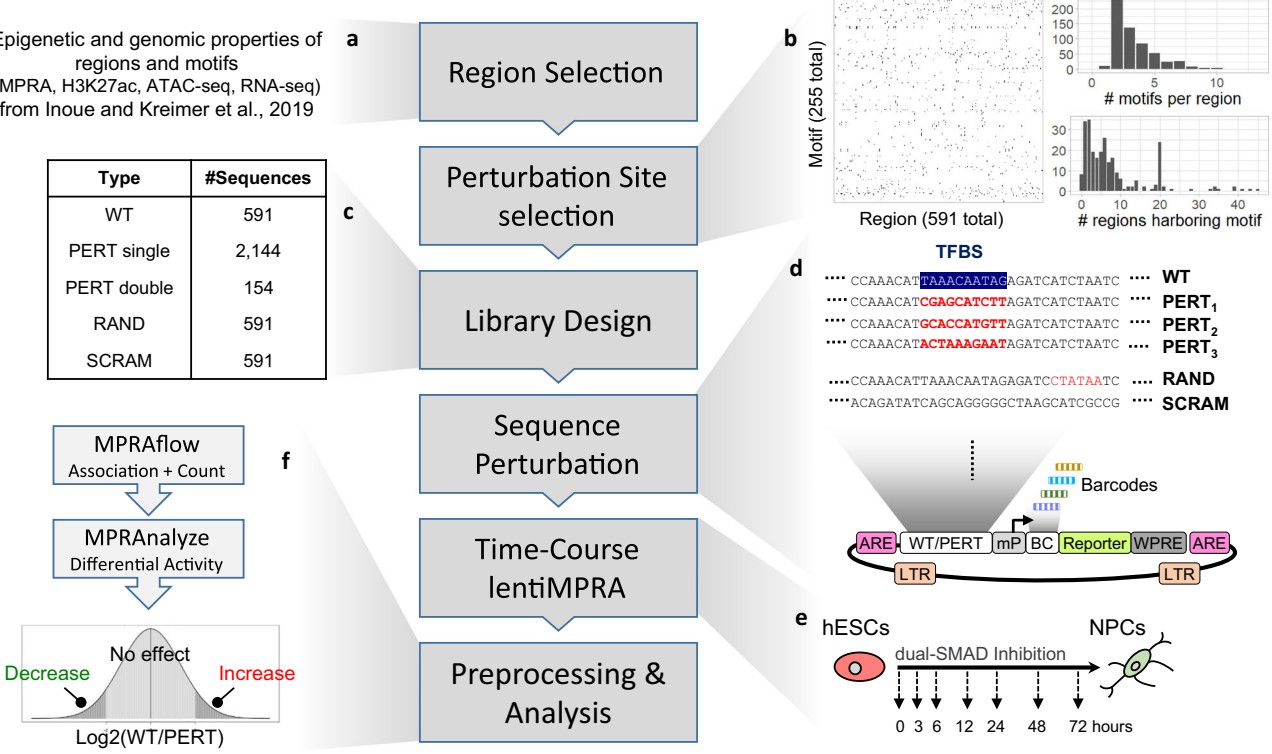

**Fig. 1 Experimental design. a** Computational framework to select regions and perturbation sites. **b** Heatmap of motif instances in the assayed regions (left); distribution of the number of motifs perturbed in each region (top right); distribution of the number of regions harboring each motif (bottom right). **c** Library design; selected regions were included in their wild-type (WT) form, selected motifs were perturbed (by altering the sequence in the predicted motif site) using three perturbation methods individually (PERT single) as well as in combination with other perturbations in selected cases (PERT double). Random sites were perturbed (RAND) and the entire WT sequence was scrambled as negative controls (SCRAM) for each WT sequence. **d** The designed sequences were synthesized and cloned into the lentiMPRA vector and associated with 15-bp barcodes. ARE antirepressor element, BC barcode. Reporter, EGFP enhanced green fluorescent protein, LTR long terminal repeat, mP minimal promoter, WPRE Woodchuck Hepatitis Virus Posttranscriptional Regulatory Element. **e** lentiMPRA libraries were infected into hESCs and following 3 days, we induced neural differentiation via dual-SMAD inhibition and obtained DNA and RNA at seven time points (0, 3, 6, 12, 24, 48, and 72 h). **f** Association between barcodes and designed sequences, and the number of barcodes observed in DNA and RNA sequencing was determined using MPRAflow[28]. Differential analysis between WT and PERT activity to determine motif regulatory effect over time was assessed using MPRAnalyze[22]. Source data are provided as a Source Data file.

and 72 h post induction;[20]). Our initial candidate set consisted of 1547 171-bp sequences that were identified as temporally active (i.e., the expression of their target barcode varied significantly, both over time and in comparison to a control sequence; "Methods"[22]). We then used FIMO[23] to computationally identify occurences of known DNA-binding motifs in each sequence (using motifs identified by Kheradpour and Kellis[24] and Weirauch et al.[25]).

Following these analyses, we chose specific regions and motifs for perturbation lentiMPRA. As we are limited by the number of sequences that can be included in a single lentiMPRA library due to low integration rate in ESCs, we developed an optimization framework to select the combination of regions and motifs that maximizes the representation of relevant genomic properties (Fig. 1a, b; "Methods"). To this end, we wanted to include regions and motifs that are associated with different temporal patterns of chromatin and gene expression signals, derived from our previous analysis of H3K27ac ChIP-seq, ATAC-seq, and RNA-seq data in the same time points[20]. We made sure to include a sufficient number of regions in which H3K27ac is induced early in the differentiation, as well as regions that gain this mark later on, and closer to the neural progenitor (NP) phase. Similarly, we selected a minimal number of motifs whose corresponding TFs are induced early in the differentiation process, as well as motifs associated with late-induced TFs. We also chose to provide explicit preference for a curated list of regions and TFs that have

been previously associated with neural induction pathways. Finally, we required that every selected motif will be perturbed in at least 20 regions (thus allowing us to observe the motif in multiple contexts), and every selected region will have at least two different perturbations (for two different motifs). With these considerations taken together, the respective experimental design problem can be represented as an optimization problem: selecting the minimal number of motif instances [(region × motif) pairs] while satisfying all of our design constraints above. In "Methods", we describe how we represent this as a connectivity problem in graphs and how we derive a solution for it using Integer Linear Programming. Applying this scheme to our data resulted in a selection of 2144 motif instances over 591 regions and 255 motifs (Fig. 1a–c).

We considered the 2144 motif instances both in their wild-type (**WT**) and in a perturbed form (**PERT**), where the sequence of the motif instance is modified in order to estimate its effect. For 100 of our genomic regions, chosen by the motifs they harbor and their importance for neural differentiation ([20]; see "Methods"), we also perturbed pairs of motifs (including two appearances of the same motif in the sequence or two different motifs), to analyze cooperative effects (Fig. 1c; "Methods"). We perturbed each of the selected motifs using three different designs that rely on two approaches (Fig. 1d): In designs 1 and 2, we identified two fixed "non-motif" sequences (i.e., sequences with minimal number of predicted motif hits—details in "Methods") and replaced the

motif with the prefix of these sequences, adjusting to the motif length. In the third design, we randomly shuffled the nucleotides of the motif ("Methods"). We also included two sets of negative controls: (1) scrambled sequences (**SCRAM**)—where we shuffle all nucleotides of each of the 591 WT sequences; (2) random sequence alterations (**RAND**) – where we randomly shuffled a small region (length of the median motif size) at a random location in each region. In total, 10,041 sequences were included in our lentiMPRA library (Fig. 1c, d).

**lentiMPRA perturbation**. The designed sequences were synthesized and cloned upstream of a minimal promoter (mP) into the lentiMPRA vector (Fig. 1d; "Methods"). During the cloning process, 15-bp random barcodes were placed in the 5'UTR of the EGFP reporter gene[26]. The association between the cloned sequences and barcodes was determined via DNA-seq ("Methods"). Lentivirus was generated and human embryonic stem cells (hESCs) were infected with the library (Fig. 1e). Following three days, to allow for viral integration and degradation of unintegrated virus, the hESC were differentiated to a neural lineage using the dual-Smad inhibition protocol[27]. Integrated DNA barcodes and transcribed RNA barcodes were quantified by DNA-seq and RNA-seq, respectively, at seven time points of neural differentiation (0, 3, 6, 12, 24, 48, and 72 h) (Fig. 1f). The library infections were carried out using three biological replicates (two replicates were infected with the same lentivirus batch, while the other replicate was infected with another lentivirus batch).

Using a computational pipeline developed in our group, MPRAflow[28], we took a stringent approach to associate barcodes with the cloned sequences. For each barcode, we required at least 80% of the reads associated with the barcode to map it to a single sequence, and a minimum of three reads supporting that assignment, resulting in over 1.4 million confidently assigned barcodes, and averaging 139 barcodes per sequence ("Methods"). We then analyzed the barcodes from the lentiMPRA infected cells and matched them with the confidently assigned barcodes of the library. Across biological replicates, we were able to confidently assign an average of 61.6% of the barcodes ("Methods" and Supplementary Fig. 1). Considering only confidently assigned barcodes that have a representation both in RNA and DNA from infected cells, we observed an average of 134.4 barcodes per sequence in each replicate (Supplementary Fig. 2), corresponding to 9948 out of the 10,041 designed sequences (2082, 2086, and 2114 sequences for perturbation methods 1–3, respectively (Supplementary Table 1)). We then used MPRAnalyze[22] to aggregate the barcodes and quantify the transcription rate induced by each tested sequence (dubbed "alpha"). We observed reproducible results between replicates (average Pearson correlation 0.98) in every timepoint (Supplementary Fig. 3), and results were highly concordant with our previously characterized lentiMPRA in the same system[20] (mean Pearson correlation 0.79, Supplementary Fig. 4). Comparing the four categories of sequences that we tested, we observe as expected, that overall, the scrambled negative controls (SCRAM) had the lowest transcriptional activity, while the unperturbed sequences (WT) had the highest (Supplementary Fig. 5). We also observed that sequences with a perturbed binding site (PERT) had a generally lower level of activity than sequences with a perturbation of random sites (RAND), confirming that perturbing known motifs have an effect larger than expected by chance. We next quantified the magnitude of deviation between PERT and WT transcription rates (Log(WT/PERT)) and compared the results between all three perturbation methods. Overall, we observed correlated results between the three methods, both in terms of the estimated transcription rate of the perturbed

sequences (average Pearson correlation 0.81) and the differential activity between the perturbed sequences and their corresponding WT sequence (Log(FC), average Pearson correlation 0.71) (Supplementary Fig. 6).

**Identification of functional TF motifs**. We next set out to identify which of the DNA-binding motifs we assayed is a functional site, i.e., a site that causes a significant change in regulatory activity when perturbed. To this end, we initially focused on sequences with a single perturbed site (rather than deletion of two sites) and used MPRAnalyze[22] to apply a set of four filters (illustrated in Fig. 2a; Supplementary Table 1a), requiring that each tested sequence passes all four filters: (1) the PERT sequence activity significantly deviates from that of the WT sequence in at least one time point (likelihood ratio test (LRT); FDR < 0.05; "Methods"); (2) the time course of PERT activity significantly deviates from that of the WT sequence (LRT; FDR < 0.05; "Methods"); (3) either the PERT (in at least one time point) or WT (in all the time points) sequences are significantly more active than the SCRAM negative controls (MAD-based z-test; FDR < 0.05; "Methods"); (4) either the PERT or the WT sequence temporal activity significantly deviate from the temporal activity observed among the SCRAM negative control sequences (LRT; FDR < 0.05; "Methods"). Overall, these filters will include sites that when perturbed cause a significant change (compared to WT) in regulatory activity in at least one time point (filter 1) and across the temporal pattern (filter 2). In addition, sequences that are potentially activating or repressing in at least one time point or across time will be included using filters 3 and 4. Our analysis will not remove constitutive sequences as long as their temporal activity is significantly different from the WT. We applied these filters to each perturbation method separately, which resulted in 747, 775, and 749 sequences in perturbation methods 1, 2, and 3, respectively (Fig. 2b, Supplementary Fig. 7, and Supplementary Table 1a). Across the three perturbation methods, we observe that most of the sequences pass all four filters and less than 10% of the sequences pass no filter, indicating that our experimental design mainly consists of functional regulatory sequences across these time points of neural induction (Supplementary Fig. 7). Comparison analysis of temporal properties from our previous work[20] confirmed that the signal of H3K27ac, ATAC-seq, MPRA, and mRNA of both the closest gene and the motif's associated TF were significantly lower (all time points combined, Wilcoxon P value $<10^{-10}$) in removed vs. retained sequences, in each of the perturbation methods, supporting our filtering approach (Supplementary Dataset 1). We observed an overall similar level of concordance between the different methods, with an average overlap of ~70% between the three methods (Fig. 2b, Supplementary Fig. 6, and Supplementary Table 1).

In the subsequent analysis, we took a conservative approach to aggregate the evidence from the three ways of perturbing motif instances. We focused on instances that had strong evidence from both approaches for perturbing a motif (i.e., random shuffle or replacement by a fixed "non-motif" sequence). To this end, we consider only instances that passed all four filters above in perturbation method 3 and in either perturbation method 1 or 2. We also require that the direction of effect (increasing or decreasing expression) to be consistent between the different methods. This resulted in 598 motif instances that had a significant and consistent effect. We refer to this set as functional regulatory sites (FRSs).

We examined the FRSs by conducting our analysis in three different axes: (i) the FRS level, i.e., perturbation of a specific motif in a specific region; (ii) the motif level, across different regions the motif appears in; (iii) the region level, taking into

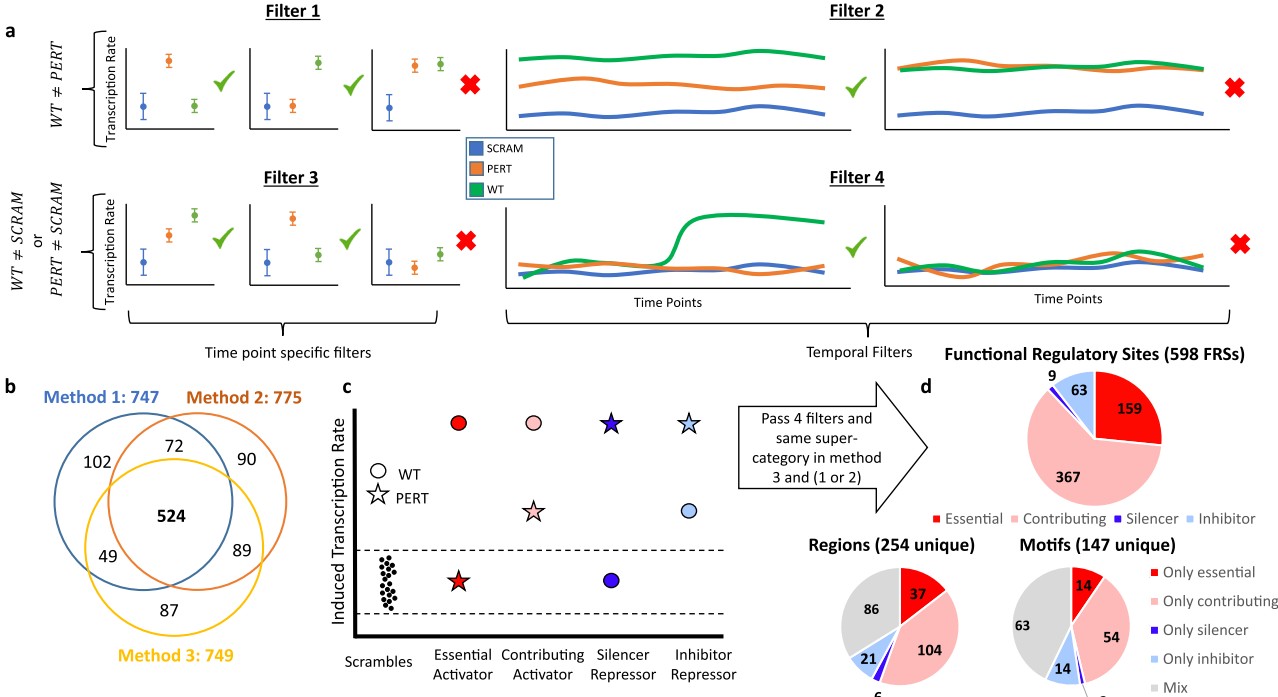

**Fig. 2 Preprocessing, consistency, and categorization of FRSs. a** Illustration of the four filters applied to perturbed sequences to remove inactive and non-functional sites, both at each timepoint and across timepoints (error bars represent mean ± 1 SD; "Methods"). **b** Number of sequences that passed all three filters for each perturbation method. **c** Definition of main and sub-categories of motif binding effects based on their effect on transcription. **d** Distribution of categories across FRSs that pass the four filters and are under the same main category (activating or dampening) in perturbation method 3 and at least one of the perturbation methods 1 or 2. The distribution is shown across FRSs (top) and across the unique motifs and regions composing the FRSs in this study (bottom).

account the various functional sites that appear in it. For each axis, we also examined how the perturbation effect may change over the different time points. While our analysis is based on the consensus set of 598 motif instances, we repeated it based on sites found by each of our three perturbation methods individually, where we observe largely consistent results (Fig. 2b, Supplementary Tables 1–5, Supplementary Dataset 1; "Methods").

**Delineating major categories of functional regulatory sites**. We first analyzed the general effect of our perturbations in all 598 FRSs. Comparing the MPRA signal of WT to PERT sequences in each time point, we generally observed a reduction in activity, indicating that perturbing the predicted motif disrupts the function of an activating TF. For a smaller portion of the sites, we observed the opposite effect, i.e., increased activity, indicating that these sequences harbor binding sites with a repressive function (Supplementary Fig. 8). Importantly, these elements do not lower the baseline transcription rate of the reporter gene, and are not transcriptional repressors, but rather reduce the expression to levels comparable to the baseline of the control sequences (SCRAM), but not below it. To avoid confusion with transcriptional repressors we refer to these elements as *dampeners*, as they dampen the activity of the enhancer. We thus divided the perturbation effects into two main categories (Fig. 2c): (1) *activators*, identified by perturbations resulting in reduced transcription (WT > PERT); (2) *dampeners*, identified by perturbations resulting in increased transcription (PERT > WT).

Out of the 598 FRSs, we observed 526 (87.9%) that had activating effects in at least one time point (and non-significant effects in the rest of the time points), and 70 (11.7%) that had dampening effects in at least one time point (and non-significant effects in the rest of the time points) (Fig. 2d, Supplementary

Table 2, and Supplementary Dataset 2), with only two FRSs alternating between activating and dampening effects at different time points (DMRTA2 motif DMRTA2_M0629_1.02 and Interferon Regulatory Factor 4 motif IRF4_M5573_1.02; Supplementary Table 2; Supplementary Dataset 2). This suggests that the direction of the effect (activating or dampening) of an FRS primarily depends on DNA sequence, and less so on the protein milieu or on other epigenetic properties that change during differentiation. Of note, as lentivirus randomly integrates into the genome, our results consider a cumulative signal from different integration locations in many cells, which essentially controls for the effects of local chromatin properties that may be present around the FRS.

To gain a better understanding of perturbation effects, we further divided our sites into four sub-categories (Fig. 2c, d, "Methods"): (1) *Essential*: activating sites that when perturbed, reduced the expression level to that of the controls (SCRAM) sequences; (2) *Contributing*: activating sites that upon perturbation reduce the expression but not to baseline levels; (3) *Inhibiting*: sites that when perturbed lead to increased activity suggesting that they encompass dampening sites that fine-tune transcription levels; (4) *Silencing*: dampening sites that block a sequence from regulating transcription, i.e., WT levels are similar to control (SCRAM) and when perturbed make the sequence active. (Fig. 2c; "Methods").

Considering this refined division, we found that 159 and 367 out of the 526 activating FRSs, correspond to categories *essential* and *contributing* respectively (Fig. 2d and Supplementary Dataset 1). Out of 72 dampening FRSs, we find 9 *silencers* and 63 *inhibitors* (Fig. 2d and Supplementary Table 1). These results represent the distribution of FRSs categories in our dataset. Notably, these FRSs are not a comprehensive list of all functional

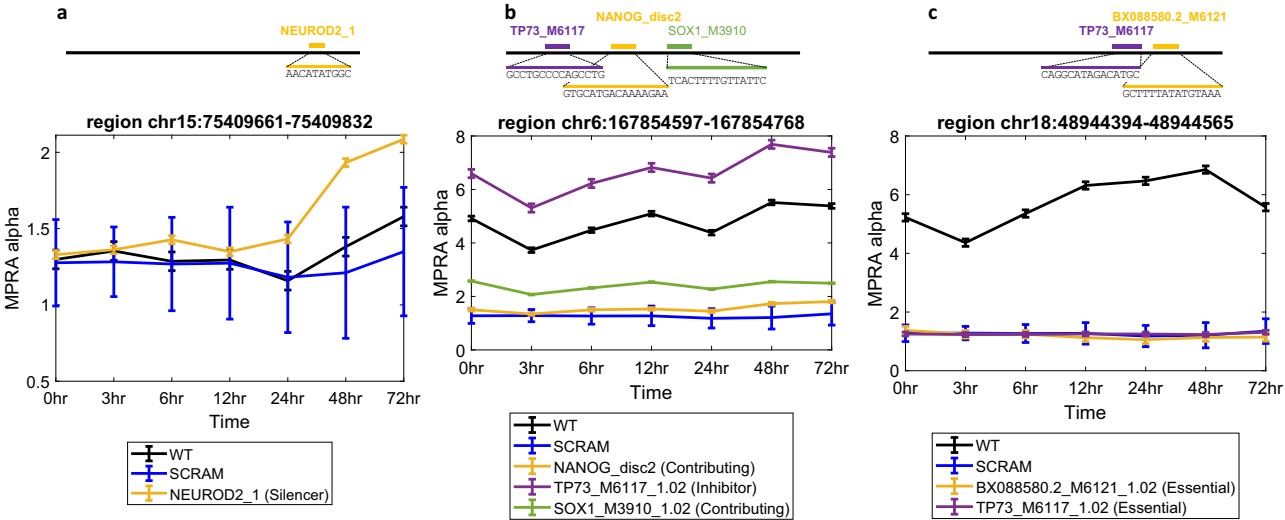

**Fig. 3 Examples of the four sub-categories of FRSs.** In each figure, the WT sequence is indicated in black, centered at the mean activity and including error bars of ±1 SD across the three replicates, and SCRAM in blue including error bars of mean ±1 SD across all scrambled sequences. Each motif is plotted in a different color, including error bars of mean ±1 SD across the three replicates, and its perturbation effect in the regions is indicated in the text box. **a** NEOROD2 has a silencer effect. **b** NANOG, TP73, SOX1 have contributing, inhibiting, contributing effects concordantly. **c** BX088580.2, TP73 both have essential effects. All genomic coordinates are hg19. Source data are provided as a Source Data file.

sites in the selected regions. For instance, we found several regions in which only dampening FRSs were identified (Fig. 2d). Since dampening FRSs only reduce the overall activating function of the region, dampener-only regions must contain additional unknown activating FRSs that were not included in our design.

We next examined how the strength of the mutation effects caused by perturbing activator sites (WT–PERT) depends on the strength of the expression generated from their respective unperturbed sequence (WT). We found that these effects scale linearly with the WT activity levels ($WT–PERT \sim a + b* WT$ for some constants $a, b$) across time. While this is trivial for essential FRSs, we found that this linear relationship still holds among contributing activators as well (median R-squared 0.95, methods, Supplementary Fig. 9a–c). When examining fold-change values, this linear relationship translates to: $FC = PERT/WT = (1-a) - (b/WT)$ for the same constants. This relationship saturates and approaches a constant $(1-a)$ for sufficiently high levels of unperturbed (WT) expression (Methods; Supplementary Fig. 9d, e). These constants therefore capture the activation dynamics of each element: $a$ determines the saturated value, and $b$ determines the rate of saturation. We observed that different FRSs within a given region often have different constants, and the same motif has different constants when harbored in different regions, suggesting that the dynamics are not context- or factor-specific, but rather a combination of both. Overall, while the relationship between WT activity and the effect of perturbation is linear, our results show that both depend on the sequence content and the specific cellular context in which it is being assayed.

**Characterization of activating and dampening motif effects.** Overall, our 598 FRSs include 147 unique motifs. Out of these 147, we observed 68 motifs that are strictly activators, 16 motifs that are strictly dampeners and 63 motifs that show either activating or dampening effects in different genomic contexts (Fig. 2d, Supplementary Figs. 10 and 11, Supplementary Table 2, and Supplementary Dataset 2). When examining the distribution of motif effects across regions (Supplementary Fig. 10 and Supplementary Dataset 2), we observe that related motifs tend to appear in the same regions and importantly— that motifs have

different, in many times opposing, effects in different regions. This is also supported by a per motif visualization showing the distribution of categories per motif (Supplementary Fig. 11 and Supplementary Dataset 2). In addition, there are groups of similar regions that contain the same motifs (Supplementary Fig. 10 and Supplementary Dataset 2). We note that most of the motifs in our dataset (~75% Supplementary Fig. 10 and Supplementary Dataset 2) appear in five or less regions. Constraining the analysis to motifs that appear in more than five regions shows that 16 out of 35 such motifs (~45%) are strictly activators and all of them have mixed effects depending on the region.

We set out to examine the aforementioned sub-categories of specific motifs (Supplementary Fig. 11). Within the activating FRSs, we observed that motifs associated with the SRY-Box Transcription Factor SOX1 are the only motifs that are enriched in the set of *essential* FRSs (i.e over-representation that is unlikely to occur by chance; hypergeometric test, FDR < 0.05; Supplementary Fig. 11). Both SOX1 and its homolog SOX2 are thought to function as pioneer factors that enable subsequent binding by other TFs[29]. This is in line with our observation that the enhancer activity is completely disrupted when these motifs are perturbed. Among the motifs that were enriched in the second category of having a contributing binding effect, we observed ZIC factors, which play important roles in neuroectoderm cell development[30].

Among the transcription factors whose motifs are associated with a silencing effect is the Neuronal Differentiation factor NEUROD2. Perturbing a NEUROD2 binding site in a late-response regulatory element (chr15:75409661–75409832 (hg19); Fig. 3a) increases the transcription induced by that sequence at the later time points (48–72 h) (Fig. 3a). While NEUROD2 is thought to be a transcriptional activator, our results are in accordance with its previously reported role as a repressor of REELIN gene expression in primary cortical neurons, by interacting with CTCF that is known to function as transcriptional repressor in a context-dependent manner[31].

Considering the set of *Inhibitor* motifs, which could fine-tune regulatory activity by partially reducing it, we saw enrichment for the P53-Like Transcription Factor TP73. For example, perturbing a TP73-binding site in region chr6:167854597–167854768 (hg19) substantially increases the activity of that enhancer across all time

points. Notably, this region also contains two functional binding sites that activate transcription, and harbor NANOG (NANOG_disc2) and SOX1 (SOX1_M3910_1.02) binding motifs (Fig. 3b). Interestingly, we also found six instances where TP73-binding motifs function as activators (Fig. 3c and Supplementary Table 1). TP73 has been shown to regulate NPC proliferation in the developing and adult mouse central nervous system[32,33] and is known to interact via its subdomains with many different partner proteins, including POU[34] which has corresponding motifs in this region and YAP1, which is known to function as both an activator or repressor[35] in a context-dependent manner[36]. These instances demonstrate that FRSs can achieve their desired transcriptional rate by combining both activating and repressive motif sites, and that using our perturbation MPRA approach allowed us to distinguish the functionality in each specific context.

When examining the distribution of sub-categories effects across motifs, we observed 84 (57%) motifs that appear in only one subcategory and 63 (43%) motifs with mixed effects (Fig. 2d and Supplementary Table 1). For most of the motifs the effects are mixed (Supplementary Fig. 11). These results suggest that enhancer activity is influenced both by the motif sequence and the surrounding sequence of the region harboring the motif (Supplementary Figs. 10 and 11).

Focusing on motifs that are consistently associated primarily with one direction of effect (activating or repressing), we next set out to analyze the effects of motifs on transcription during our time course, by aggregating the results from all their respective instances. We summarized the signals of motifs that show activating or repressing cumulative effect (Fig. 4). Among the TFs associated with activator motifs, we observe the neural markers SOX, LHX, ZIC, and FOX families[20,29,30,37–44] (Fig. 4a), as well as motifs associated with factors known to be involved in neural induction, such as OTX2[45,46] and PAX6[40,47]. Consistent with our recent characterization of neural induction associated TFs[20], we also identified Iroquois Homeobox Protein 3 (IRX3) to be one of the strongest activating motifs. Among the TFs associated with repressive activity (Fig. 4b), we observed factors from the HOXD gene family, which are thought to function as repressors when bound in monomeric form[48]. We also found an enrichment for a SIN3A motif, which is generally known to interact[37] with histone deacetylase (HDAC) and function as a transcriptional co-repressor[49]. It was also reported that the SIN3A/HDAC co-repressor complex was involved in the maintenance of ESC pluripotency[49,50].

To examine how the effects of motifs change over time, we clustered the signal of all activating and repressing motifs. We observed that the magnitude of effects often changes over time in a manner proportional to the unperturbed expression level (Supplementary Fig. 9). These effects range from perturbations that are effective only at the ESC stage to those that influence late-induced regions (Fig. 4). Enrichment analysis of the TFs (both activating and repressing) in the early cluster (Fig. 4a, b) indicated their involvement in processes related to cell differentiation, cell fate commitment, and regulation of development for the top ten categories, whereas enrichment of late response TFs (Fig. 4a, b) indicated, more specifically, categories related to neurogenesis and nervous system development[51]. These results support the functionality of these clusters in earlier and later stages of neural differentiation. For example, enhancers that have OTX2-binding sites reach their peak activity during the neural progenitor cell (NPC) stage. When the OTX2 sites are perturbed, the activity at later time points (48–72 h) was decreased (Fig. 4a, c). Similarly, NPC enhancers harboring IRX2/3 (Fig. 4a) or BARHL1 (Fig. 4d) motifs decreased in activity when the binding sequences were mutated. Correspondingly, we observe that OTX2, IRX2/

3, and BARHL1 mRNA levels peak at later time points (48–72 h) (based on data published in ref. [20]). When HOXD sites (HOXD12_M5560_1.02, HOXD9_2) were mutated, the activity at later time points (48–72 h) (Fig. 4b, e, Supplementary Table 2, and Supplementary Dataset 2) was increased. These findings indicate that these binding sites have different levels of induced activity at distinct time points of neural differentiation. This suggests that the abundance of the binding TF (i.e., the TF's mRNA levels) at a given time point and the abundance of additional cell-state specific factors (e.g., expression of other TFs) play a significant role in proper enhancer activity.

Interestingly, we also observed TFs whose corresponding motifs show both activating and repressing effects in different regions (Fig. 4a, b, Supplementary Dataset 2, and Supplementary Figs. 10 and 11). For example, different motifs for the Zinc finger protein (ZIC) family have repressing and activating effects across different regions (ZIC2 and ZIC3). Members of the ZIC family are involved in neurogenesis and are known to function as both transcriptional activators and repressors in a context-dependent manner during embryogenesis[52]. In addition, we observed both effects for the ZEB1 motif in different regions (Fig. 4a, b and Supplementary Dataset 2) in concordance with the role of ZEB1, acting as both a transcriptional activator and repressor during neurogenesis[53,54]. We saw similar effects for the RARG motif (Fig. 4a, b and Supplementary Dataset 2). RARG is a retinoic acid receptor (RAR), a family of factors that plays a role in developmental processes and acts as a ligand-dependent transcriptional regulator. When bound to ligands, RARs activate transcription, whereas in their unbound form they repress transcription of their target genes[55].

### Perturbation of motif pairs identifies different modes of motif interaction.

We next examined the activity of the assayed regions as composite functional units consisting of multiple FRSs. Our 598 FRSs include 254 unique genomic regions. We observe complexity in these regions in terms of having sites with different direction of effect and different sub-categorization (Supplementary Fig. 10). Specifically, when examining the set of significant perturbation effects in those regions, we observed 141 cases (~56%) with only activating effects (Fig. 2d, Supplementary Dateset 2, and Supplementary Fig. 10), which is consistent with our analysis being focused on regions that were previously identified as enhancers during neural induction[20]. We found 86 regions (>30%) that harbor both activating and repressing motif instances. This suggests that regulatory activity within these enhancers can be achieved by fine-tuning of binding effects, including both activating and repressing motifs to achieve the desired regulatory function. This phenomenon of context-dependent repression by transcriptional activators is consistent with what was previously reported in yeast[56], drosophila[57], and mammalian cells[58]. Regions with multiple *essential* FRSs, all required for regulatory activity, offer support to the "enhanceosome model" of a specific combination of factors being required in an all-or-nothing machinery[15]. In contrast, regions with multiple *contributing* FRSs supports the "billboard model", of a flexible modular machinery that fine tunes the induced transcription levels by having independently contributing factors[59]. These results demonstrate that different regulatory sequences may be governed by either the enhanceosome or the billboard model, and some appear to be governed by a combination of both.

We wanted to further examine how pairs of motifs interact in regulatory sequences. To that end, we examined the results of perturbing pairs of motifs, both individually and in combination, to determine how different binding sites interact in a single region

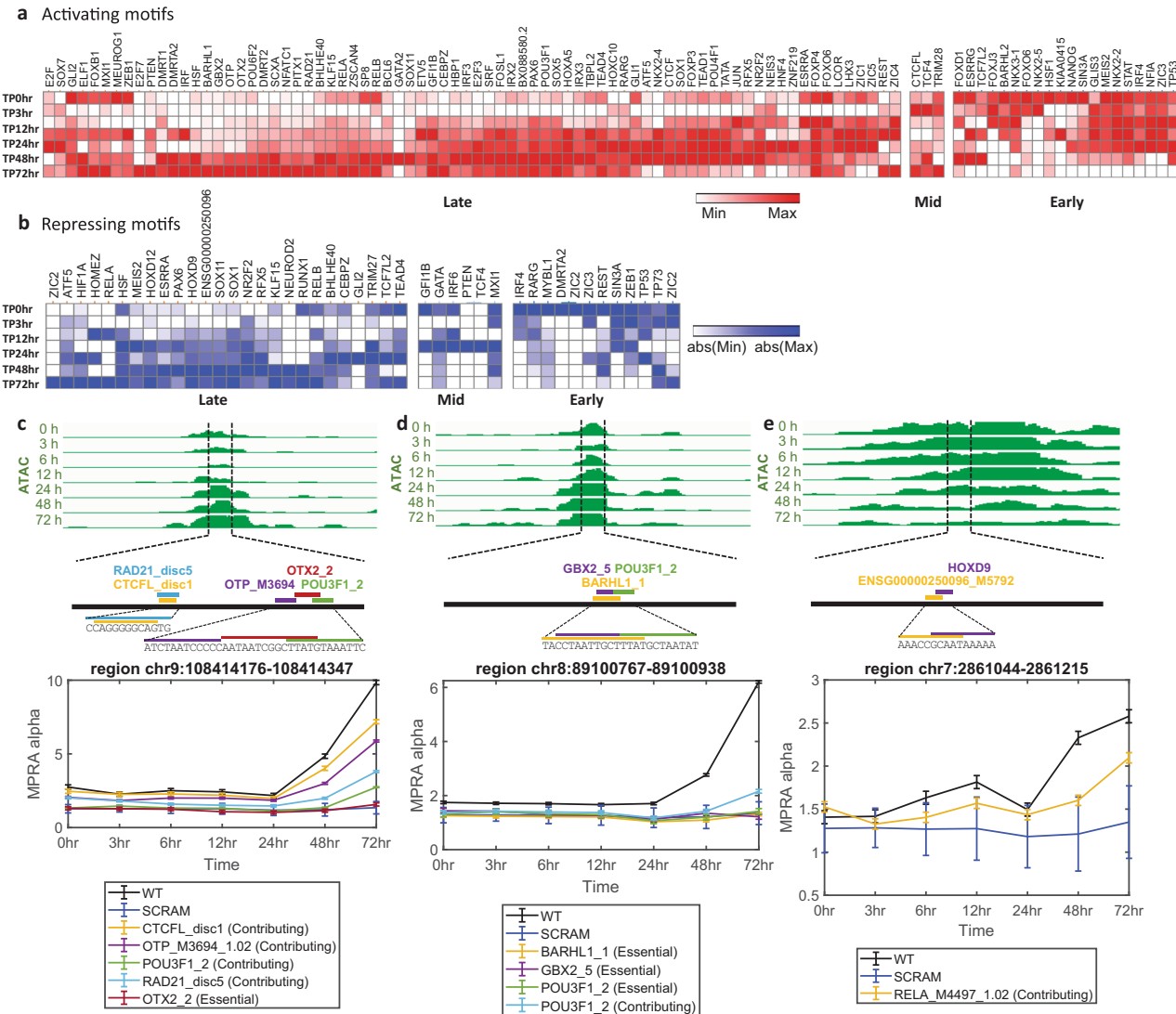

**Fig. 4 Temporal motif effects.** The **a** activating or **b** repressing motifs in at least one time point. red—activator motifs, blue—repressor motifs. Color scale indicates the average of the perturbation signal (LogFC(WT/PERT)) across all significant instances of motifs for a specific TF (row normalized). Data are organized using hierarchical clustering and early, mid, and late clusters are indicated. Genome browser snapshots of assaysed sequences near predicted sequence motifs that are associated with **c** *OTX2* and **d** *BARHL1* TFs, showing the motifs that were perturbed and their effect on activity across time points. **e** HOXD9 repressor motif example. Line plots similar to Fig. 3, mean activity ± 1 SD across the three replicates. All coordinates are hg19. Source data are provided as a Source Data file.

(Fig. 5a and Supplementary Dataset 3). We considered the FRSs to have independent effects (following the billboard model) if the effects were log-additive: perturbing both sites was equivalent to multiplying the effects of perturbing each site separately. We used MPRAnalyze[22] to test this hypothesis for each assayed pair in each perturbation method, by including an interaction term in the model that captures the effect of perturbing both sites while accounting for the effect of perturbing both sites individually ("Methods"). We considered pairs to have significant interaction if the size of the interaction term was larger than 0.5 and the test was statistically significant (BH-corrected $P < 0.05$). We then defined interaction as "consistent" if the pair were labeled the same (either significant or non-significant) in perturbation methods 3 and either 1 or 2, and removed inconsistent pairs from the analysis. We removed pairs in which none of the perturbations are functional, by requiring that at least one of the single perturbations pass the filtering scheme we described above. Finally, to make interpreting the results easier, we excluded pairs in which the assayed sites overlap since overlapping sites cannot

be conclusively independent. Overall, out of 149 examined pairs, 24 pairs remained, of which 13 were log-additive, consistent with a billboard model of cooperation, and 11 had significant non-additive interactions (Fig. 5b). While the small number of functional pairs in our results does not allow for extensive or systemic analyses, we do find anecdotal evidence of different cooperation models operating in different regions.

Among the billboard-consistent pairs, we found chr10:1002 06539–100206710 (hg19), residing in an intron of the *HPS1* gene, contains two FRSs each containing a motif instance of ELF1 (ELF1_known3), a transcription factor known for its binding near prefrontal cortex splicing QTL SNPs[60] and for its role in brain development[61]. Both FRSs are activators, but do not have an identical effect, with one driving down transcription to SCRAM levels when perturbed (*essential*), and the other having a milder effect (*contributing*). Perturbing both FRSs in this region further reduces the expression to levels significantly below the SCRAM baseline (Fig. 5c). In addition, we find that additive contribution can also apply to cooperating activators and dampeners, as in

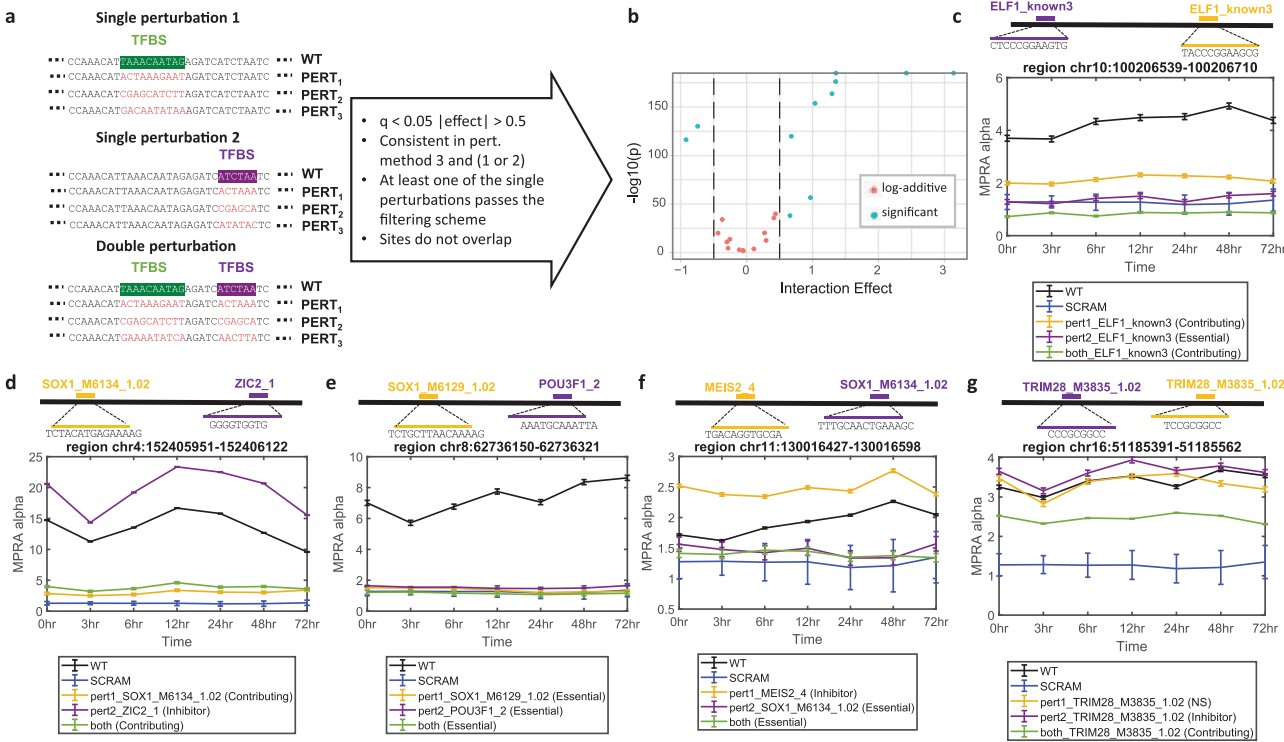

**Fig. 5 Double perturbation scheme. a** Experimental design for perturbing two single motifs separately and then a double perturbation of both simultaneously, and the requirements for being included in downstream analysis. **b** Volcano plot for the model testing for log-additivity of the individual effects. **c–g** Examples of double perturbation results demonstrating different patterns of cooperation: log-additive effects consistent with a billboard model (**c**); log-additive effects of one dampening and one activating element (**d**); fully dependent cooperation consistent with the enhanceosome model (**e**); a billboard-enhanceosome hybrid model with one required element and one with a dampening effect (**f**); a redundancy example, perturbing either motif has negligible effect, but perturbing both has a substantial effect (**g**). All coordinates are hg19. Line plots similar to Fig. 3, mean activity ± 1 SD across the three replicates. Source data are provided as a Source Data file.

chr4:152405951–152406122 (hg19), an intronic region in the FAM106A1 gene body, which contains an FRS with a SOX1 motif that has a contributing effect and an FRS with a ZIC2 motif that has an inhibiting effect. Perturbing both sites results in an additive effect: transcription levels that are lower than WT, but higher than those obtained when perturbing the SOX1 motif alone (Fig. 5d). In the non-additive regions, we found both enhanceosome and composite examples. In the all-or-nothing enhanceosome model, different elements act in a fully dependent manner. For example, chr8:62736150–62736321 (hg19) contains two essential functional sites: a SOX1 (SOX1_M6129_1.02) and a POU3F1 (POU3F1_2) motif, both necessary for activity. Perturbing either one, and concordantly both, reduces induced transcription to SCRAM levels (Fig. 5e). Both factors are known to have a key role in determining neural fate[62]. In a combination of the billboard and enhanceosome models, some factors are required for any activity while others are independently contributing. For instance, chr11:130016427–130016598 (hg19), downstream of the APLP2 gene which is involved in neural differentiation[63] contains two FRSs: a dampening site with a motif for neural factor MEIS2, and an essential FRS harboring a SOX1 motif. Perturbing both sites results in a reduction of activity to the SCRAM levels, indicating that the SOX1 FRS is required for the overall activity of the region, whereas the dampening MEIS2 FRS is only functional in the presence of a functional activator (Fig. 5f).

In addition, we found regions that follow neither the billboard or enhanceosome models. In chr16:51185391-51185562 (hg19), upstream of the promoter of neurogenesis regulator SALL1[64], we find two binding sites of TRIM28. When perturbed individually,

one site has no effect, while the other has a mild dampening effect. However, when both are perturbed the effect is a significant decrease in activity. This potentially demonstrates a redundancy mechanism, whereby either binding site is sufficient for the WT activity, and both need to be perturbed in order to disrupt it (Fig. 5g). Overall our results demonstrate the power and potential of perturbation MPRA in uncovering a variety of different patterns of interaction and elucidating the complex regulatory grammar governing these behaviors.

## Discussion

Regulatory elements play a major role in cell-type-specific response to environmental conditions and perturbations. Teasing out the regulatory rules and sequences responsible for these responses could lead to a better understanding of how variations in these sequences alter their activity, and allow the accurate design or targeting of specific sequences for therapeutic purposes. Here, we used perturbation MPRA across seven time points of neural differentiation to characterize the regulatory grammar during early stages of neural induction. Our work allowed us to evaluate the effect of intact motif instances over time and annotate these instances into four major categories (*essential, contributing, inhibiting,* or *silencing*). We observe that generally a FRS either has an activating or repressive effect across all time points, suggesting that the binding motif and surrounding region largely determine the direction of effect, and that the magnitude of this effect changes over time, in a manner proportional to the activity of the WT sequence, in different cellular environments, indicating earlier and later functional motifs in this process. Finally, by carrying out two motif perturbations in a single

sequence, we observed different modes of interaction between pairs of motifs.

Several studies have utilized MPRAs to characterize how TF binding may affect regulatory activity. However, these studies examined a small number of TF motifs and assessed their functional effects in a limited number of conditions or cell types. For example, placing TFBSs at different numbers, order, spacing and orientation on 'neutral' background sequences allowed the dissection of regulatory grammar in a human hepatocellular carcinoma cell line[59]. One common finding is that the number of TFBSs (i.e., homotypic clusters of TFBSs[65]) largely determines expression and this relationship follows a non-linear increase with an eventual plateauing of expression[56,59,66–68]. Grossman et al.[19] used both synthetic and endogenous sequences to specifically test the effect of PPARγ binding motifs and show that distinct sets of features govern PPARγ binding vs. enhancer activity. Specifically, they found that PPARγ binding is largely governed by the affinity of the specific motif binding site while the enhancer activity of PPARγ binding sites depends on varying contributions from dozens of TFs in the immediate vicinity, including interactions between combinations of these TFs. Kheradpour et al.[9] examined five predicted TF activators and two predicted repressors and measured effects of their motif disruption in regulatory elements using MPRA. Their findings indicate that disrupting predicted activator motifs abolishes enhancer function, while changes in repressors maintain enhancer activity. They point to evolutionary conservation, nucleosome exclusion, binding of other factors, and motif affinity, as being predictive features of enhancer activity.

Here, we analyzed the effect of over 250 motifs with three different perturbations using two approaches. In the first approach, we replaced the motif with two different "non-motif" sequences and in the second approach, we scrambled the motif's nucleotides. All these perturbations showed high reproducibility between replicates ($r > 0.95$). Analyzing and comparing the three perturbation methods, we observed a similar level of overlap between the different methods, but we do not observe more consistency between perturbation methods 1 and 2 than either one is with perturbation method 3 (Fig. 2b, d, Supplementary Fig. 6, and Supplementary Tables 1–5). This may indicate that at least one of the fixed-sequence perturbation methods potentially introduces bias that separates it from the other, e.g., by forming de novo binding sites with endogenous sequences adjacent to the perturbed sites. Since methods 1 and 2 insert a fixed sequence, this introduced bias could be systemic across the assayed regions and skew downstream results. For future experimental designs, we suggest using a more robust perturbation approach that randomly shuffles the nucleotides of the perturbed site and is less likely to introduce systemic biases.

We cataloged the function of 598 FRSs representing 254 unique endogenous regions and 147 unique motifs. Approximately 90% of FRSs act as activators with ~30% of them as essential and the rest as contributors. This finding is also in line with a saturation-based MPRA that analyzed ten disease-associated promoters and ten enhancers, finding that the majority of mutations lead to a reduction in activity (i.e., act as activators that when mutated reduce activity)[69]. In addition, while our data do not contain FRSs that repress transcription below the baseline rate, we found many instances of binding sites that have a repressive effect on the function of the enhancer itself: reducing the level of induced transcription, or even completely blocking the enhancer's activity. These instances suggest that enhancers can be kept in a pseudo-poised state: residing in open chromatin but being blocked from activity by TF binding, and that repressive factors are often bound to

functional enhancers as a mechanism for fine-tuning transcription levels.

Finally, a smaller subset of sequences was perturbed in two locations, where we perturbed two single motifs separately and jointly to assess their interaction, as a proof of concept (Fig. 5). To model these interactions, we used the billboard model of independent contribution as our null hypothesis, by assuming that the effect of each individual contribution is log-additive[19]. We tested this hypothesis using MPRAnalyze[22] for each pair in each perturbation method ("Methods"). Only pairs which showed consistency (in perturbation methods 3 and either 1 or 2) in the significance of their interaction term (determined by the magnitude and $P$ value; "Methods"), where the single motifs were not overlapping, and at least one of the single perturbation is a FRS, were considered further in our analysis. Overall, out of 149 examined pairs, 24 pairs remained, of which 13 were log-additive (Fig. 5b–d), consistent with a billboard model of cooperation, and 11 had significant non-additive interactions (Fig. 5b). In the latter category, we observed different TF cooperation models, including the "enhanceosome model" in which a strict composition of TFs are required for an enhancer's function (Fig. 5e), a hybrid of billboard and enhanceosome models (Fig. 5f) in the same region and instances that do not fall under any of these categories (Fig. 5g). Notably, for FRSs containing two instances of the same motif, the single perturbations did not have identical effects, consistent with the growing body of work showing that the function of an enhancer depends on the specific locations and distances between binding sites, and not only of their presence[56,59,66–68]. Albeit being underpowered in the number of functional pairs does not allow for systematic conclusions, our anecdotal examples demonstrate the complexity of different TF cooperation models.

Examining whether we can gain a better understanding on the determinants of timepoint-specific regulatory activity using this model system, revealed complex results, suggesting that motif sequence alone is less likely to determine temporality without the context of the surrounding region and other bound factors (Supplementary Note 1, Supplementary Fig. 12, and Supplementary Dataset 4, 5, and 6). Therefore, future challenges following our work will include developing strategies to further understand regulatory logic and its determinants across different conditions. For example, using endogenous manipulations via CRISPR to examine the function of specific motifs and their combinations across different cellular conditions.

To address whether temporal activity of the functional regulatory sites (FRSs) are consistent with TF temporal binding using the following three strategies: first, we used RNA-seq data from[20,70] to compare the timing of motif importance with the respective TF expression. Testing this correlation did not show conclusive results. We speculate that this is due to the nature of our analysis which is motif-based, and since similar sequence motifs are not independent, it is likely that the annotation of the FRSs suffers from misclassification of the binding factor. In addition, even if the exact factor was known, it is not established in current literature that the magnitude of TF gene expression is directly correlated with its regulatory effect, so a strong correlation is not necessarily expected. Second, we examined the overlap of ChIP-seq peaks of different TFs in hESC-derived neuroectoderm[71] with regions where SOX1 motifs were perturbed, for sufficient statistical power. We observe significant overlap (Fisher exact test FDR < 0.05) of ChIP-seq peaks of OTX2 and SOX2 factors for FRSs compared to regions that were filtered out using the four filters described previously. This indicates that the signal we are observing using perturbation MPRA is consistent with endogenous binding of the key transcription factors that play pivotal roles in ES-to-neural differentiation[39,70]. Finally,

we utilized the data we collected in our previous work[20,70] of RNA-seq following overexpression of theseTFs: BARHL1, IRX3, LHX5, OTX1/2, PAX6. For the FRSs that contain motifs of these factors, we observe that ~85% of their closest genes are differentially expressed (compared to hESC; FDR < 0.05). This serves as an additional support of the endogenous functionality of motifs of these factors in these regions. Comparing the number of differentially expressed genes that are closest to the FRS to the distribution of the total number of differentially expressed genes, for each overexpressed TF, yielded a statistically significant result for PAX6 (Fisher exact test $P$ value <0.02). However, a larger number of tested FRSs will be needed to make more rigorous conclusions.

During early neural induction, pluripotency-associated genes are rapidly downregulated and neural-associated genes are induced by a variety of factors[27,40]. As such, the rapid differentiation of hESCs into neural cells provides an exceptional model to study motif effects and how they change across developmental time points. Using this model, we previously interrogated[20] the temporal dynamics of gene expression (RNA-seq) and gene regulation (ATAC-seq, H3K27ac and H3K27me3 ChIP-seq and lentiMPRA) at seven time points during early neural differentiation. Our current work further validated the motifs and TFs identified in our previous report to have temporal effects across neural induction. For example, we find that FRSs harboring BARHL1 and IRX3 motifs exhibit time point-specific activating effects and show changes in magnitude over time, with higher signal at the NPC state—supporting their suggested role in neural induction (Fig. 4).

Overall, our results provide an atlas of motif function across early time points of neural differentiation by directly testing hundreds of regulatory regions for the function of the motifs they harbor. To the best of our knowledge, this provides the first comprehensive perturbation MPRA study across a developmental time course, showing clear changes in regulatory activity over time. This system provides a model for how perturbation MPRA can be leveraged to identify and characterize in a high-throughput manner the functional effects of regulatory sequences across different cellular conditions/perturbations.

## Methods
### Computational analysis
*Perturbation MPRA library design*. Choosing region and motif combinations: General description: Our previous analysis[20] points to a large number of regulatory regions of interest as well as multiple motif hits within those regions. Our goal is to select the most informative set of [region × motif] combinations (each corresponding to a motif instance) so as to fit within a single MPRA design. To address this, we developed a selection scheme to represent various biological aspects of our system and account for experimental limitations for the number of assyed sequences.

To do this, we formalize the information that we have about the motifs and regions as a tripartite graph, with one layer of nodes corresponding to DNA regions, another layer of nodes that represent motifs and a third layer of nodes, each representing a different property of motifs or regions (Supplementary Fig. 13). The region layer consists of the 1547 genomic regions we identified in our previous work[20] that show temporal activity when tested using lentiMPRA in the same seven time points. The motif layer consists of motif hits found in those regions computationally (using Fimo ($P$ value <10^-5; Grant et al.[23]) with two sets of TF motifs[24,25]). Edges between the first two layers connect every motif with the regions in which it occurs. Each node in the third layer corresponds to a property of interest which characterizes a subset of the motifs and regions that are represented in the first two layers. These properties are based on genomics assays from our previous work[20] (based on ATAC-seq, H3K27ac and H3K27me3 ChIP-seq and RNA-seq data from these seven time points). For instance, we identified several temporal patterns associated with each data modality and designated each of these patterns as a node (e.g., a node for "regions that have a transient peak in H3K27ac 48 h of post induction"). We then connect a region to a node if that respective pattern is observed in that region in the endogenous genome. Similarly, we connect a motif node to a property node. For example, a node for "motifs with an associated TF that is expressed 24 h post induction". We then connect a motif to a node if that respective pattern is observed for that motif in the endogenous genome. We

describe the "property layer" and its edges with the "motif" and "region" layers in greater detail below.

Altogether our graph now has 1547 region nodes, 4393 motif nodes and 68 property nodes. These nodes are connected by a total of 99,165 edges. Our goal now becomes to find the minimum number of [region × motif] combinations (each representing a specific motif instance, or—equivalently—an edge in our graph) that will guarantee a sufficient coverage of each property. In other words, we want to select a minimal number of motif-region pairs such that every "property node" in our third layer is connected by an edge to a sufficient number of motifs and regions (as detailed below). Having staged our data in a tripartite graph allowed us to re-state our goal as a constrained optimization problem-guaranteeing minimal level of connectivity for the third layer, while minimizing the number of selected nodes and edges in the first two layers. Since this problem is NP- hard, we followed the common practice and formulated it as an integer linear program (ILP), which can be solved efficiently through a range of heuristics with available solvers. With this ILP, we were able to select 591 regulatory regions and 255 motifs that are organized into 2144 region-motif pairs. Below, we provide a more in-depth description of this process.

*Defining the property layer:* We composed a list of biological properties based on published literature and on ATAC-seq, H3K27ac and H3K27me3 ChIP-seq and RNA-seq data we produced and analyzed in our previous paper[20]. The biological properties of TFs associated with motifs and regions include: (i) TF/region is induced/active at a specific time point. (ii) TF/region binds/belongs to significantly overlapping sub-clusters (as defined in[20]) of temporal MPRA and H3K27ac/ATAC-seq/RNA-seq signals. (iii) The TF/the proximal gene for the region is a known neural factor or belongs to one of the pathways defined below. Known neural factors: POU3F1, MYT1L, SOX2, POU3F2, LHX2, PAX6, ASCL1, SOX1, OTX2, ZNF521, NEUROG1, NEUROG2, NEUROG3, NEUROD1, NEUROD2. Pathways taken from KEGG[72]: FGF/MAPK signaling pathway hsa04010, IGF-1/mTOR signaling pathway hsa04150, Wnt/Ca + /PCP signaling pathway hsa04310, Sonic Hedgehog signaling pathway hsa04340. (iv) Hand-picked TFs (POU3F1, POU3F2, SOX2, SOX1, PAX6, OTX2, LHX2, NEUROG1, NEUROG2, NEUROD2, SP8, IRX3, SOX10, PKNOX2, HHEX, LMX1A, BARHL1, LHX5, NR2F2, DMBX1, MEIS2, OTX1, SOX21, FOXB1, SOX5, MEIS3, HOMEZ, TCF3, TCF4, ZIC1, ZIC2, ZIC3, ZIC4, ZIC5), including factors known to have a role in neural differentiation based on previous literature[20,38–44], or based on their expression in neuroectoderm in mouse embryo, or show high "TF activity score" in the relevant time points in our data[20]. The direct edges from motifs and regions to properties, represent the biological properties a region or a motif satisfies as described above.

*The optimization program*:

1. Minimize: $(\sum_{r \in R} \theta_r) + 3 * (\sum_{(t,r) \in E; t \in T; r \in R} e_{t,r})$
   Subject to:
2. $\sum_{(t,p') \in E; t \in T} \theta_t \geq 12$ $\quad\quad \forall p' \in P$
3. $\sum_{(r,p') \in E; r \in R} \theta_r \geq \min\{17, deg_R(p')\}$ $\quad \forall p' \in P$
4. $\sum_{(t,r') \in E; t \in T} \theta_t \geq \theta_{r'} \min\{3, deg_T(r')\}$ $\quad \forall r' \in R$
5. $\sum_{(t',r) \in E; r \in R} \theta_r \geq \theta_{t'} \min\{20, deg_R(t')\}$ $\quad \forall t' \in T$
6. $e_{t,r} \geq \theta_t + \theta_r - 1$ $\quad\quad \forall(t,r) \in E; t \in T; r \in R$
7. $\sum_{t \in T_i} \theta_t \leq 2$ $\quad\quad\quad \forall T_i$
8. $\sum_{t \in T_i} \theta_t \geq 1$ $\quad\quad\quad \forall T_i \in HandPicked$
9. $\sum_{r \in R} \theta_r \geq 0.4 * |R|$
10. $\sum_{(t,r) \in E; t \in T; r \in R} e_{t,r} \geq 5 * \sum_{t \in E_p} e_{t,r}$
11. $\sum_{t \in T_B} \theta_t \geq 1.5 * \sum_{t \in T_s} \theta_t$
12. $\theta_t, \theta_r, e_{t,r} \in \{0,1\}'s$

*The decision variables represent the following*: $\theta_t$ is a binary variable that indicates whether we chose the motif $t$; $\theta_r$ is a binary variable that represents whether the region $r$ was selected. $e_{t,r}$ is a binary variable that denotes whether a motif × region pair ($t$ and $r$) has been selected.

*Parameters include*:
P—represent the properties.
R—represent the regions.
T—represents the motifs.
deg_R(p)—represents the number of edges connecting property p to regions.
deg_R(t)—represents the number of edges connecting motif t to regions.
deg_T(r)—represents the number of edges connecting region r to motifs.
$T_i$ is a subset of $T$ that contains all the motifs corresponding to TF i.
$E_p$ as a subset of the edges with lower confidence (i.e., edges that connect to properties representing non significantly overlapping sub-clusters of temporal MPRA and H3K27ac/ATAC-seq/RNA-seq signals),
We define $T_B$ as the subset of motifs connected to at least 5 regions, and $T_S$ as the subset of motifs connected to fewer than five regions.

*Constraints:* The constraints described in the equations above ensure that: (1) Each property is connected to at least 12 motifs. (2) Each property is connected to at least 17 regions (or all regions if it's below 17). (3) Each region is connected to at least 3 motifs. (4) Each motif is connected to at least 20 regions. (5) An edge is active if both nodes of the edge are active. (6) For each TF, no more than two motifs are chosen. (7) All hand-picked TFs are used at least once. (8) At least 40% of all regions are used. (9) At most 1/6 of the total edges used are low confidence edges. (10) At least 60% of motifs chosen are motifs connected with many regions ($T_B$), s.t. the solver does not bias towards lowly connected motifs. (11) All variables are binary.

For each $T_i \in$ *Hand-picked*, one representative motif must be in the solution.

*Our objective* is to minimize the overall number of MPRA sequences to design. It is a sum that accounts for corresponding to the number of unperturbed (WT) regions plus the number of perturbations (i.e., regions and motif combinations). We multiply by 3 since we have three perturbation methods (i.e., we need three MPRA sequences for every pair).

Different categories of sequences designed on the array: Overall, the solver picked 591 regions, 255 unique motifs which correspond to 166 unique TFs. We used the combinations of region and motifs chosen by the solver to represent the following sequence categories on the array (Supplementary File 1):

1. One motif is perturbed in the sequence. For combinations of regions and motifs where the motif is detected once in the sequence (**hit1** $N = 1620$).
2. Two motifs of the same motif are perturbed in the sequence. For combinations of regions and motifs where the motif is detected twice in the sequence: if the $+/-$ strand carry exactly the same motif we only replace the motif one time in the $+$ strand (**hit2** $N = 62$), otherwise (**hit2diff** $N = 90$) we perturbed each motif separately and then both of them—starting with the $+$ strand. If three or more hits of the same motif are observed – we discard those region-motif combinations ($N = 52$).

Additional to the combinations picked by the solver, we considered the 591 WT regions and added more combinations (not chosen by the solver) that contain motifs of the following 11 TFs. These TFs were chosen (LHX5, MEIS2, PAX6, FOXB1, SOX1, IRX3, OTX2, ZIC2, SP8, POU3F1, HOMEZ) based on their high "TF activity score" in the relevant time points in our data[20] and their mRNA expression in neuroectoderm in the mouse embryo.

3. One motif is perturbed in the sequence. For combinations of region and motif where the motif is detected once in the sequence (**Overexpressed_hit1** $N = 221$ and **Overexpressed_permutation** $N = 58$).
4. Two motifs of the same motif are perturbed in the sequence. For combinations of regions and motifs where the motif is detected twice in the sequence: if the $+/-$ strand carry exactly the same motif we only replace the motif one time in the $+$ strand (**Overexpressed_hit2** $N = 3$) otherwise (**Overexpressed_hit2diff** $N = 1$) we perturbed each motif separately and then both of them $-$ starting with the $+$ strand.
5. Combinations of two or more motifs are perturbed in the sequence. For combinations of regions and motifs where we observe two or more different motifs in the sequence (**Overexpressed_permutation** $N = 125$). We examined combinations of motif hits of these 11 TFs in our regions.

Overall, most of the data include a single motif perturbation per region ($N = 2144$) and a smaller part with two or more motif perturbations per region ($N = 216$ out of those: $N = 154$ two motifs; $N = 62$ more than two motifs) comprising a total of 2360 designed region and motif sequences.

We also assayed WT and control sequences:

1. We assayed 591 **WT** sequences. WT sequences are the endogenous 171-bp sequences.
2. We assayed 591 scrambled sequences (**SCRAM**). Scrambled sequences are based on WT sequences with shuffled nucleotides, creating a set of negative controls.
3. We assayed 591 sequences with random alterations (**RAND**)—where we randomly chose a location in the region and perturbed the median motif size (12 bp) starting in that location, creating an additional set of negative controls.

We perturbed predicted motifs within each genomic region (2360 combinations) using three perturbation approaches: the first two replace the predicted binding site with a "non-motif" sequence whereas the third one shuffles the nucleotides of the predicted binding site described in the next section. For the **RAND** sequence category, we used the same three perturbation approaches.

Different motif scrambling (perturbation) approaches

Approach 1—create "non-motif" sequences following these steps:

1. Use all the 2464 MPRA sequences we designed in our previous work[20] based on their potential to be active during neural differentiation.
2. Count #di-nucleotides and calculate their percentage of appearance in those sequences.
3. Create a di-nucleotide scrambled sequence in the length of the maximal motif, i.e., "scrambled motif".
4. Create 1000 maximal length "scrambled motifs".
5. Run these 1000 "scrambled motifs" through Fimo[23] and choose the ones with the lowest number of motif hits ($P$ value $<10^{-4}$) $-$ 13 "scrambled motifs" had 0 hits.
6. In each chosen combination of region and motif (described in the previous section)—replace the motif appearance with the prefix of the "scrambled motif" (adjusting to each motif length) using these two strategies that avoid motifs creation in the edges of the sequences: (1) use 3 bp downstream and upstream of the motif in the original sequence (2) use the original sequence. Repeat this 13 times using each one of the "scrambled motifs".

7. Run the sequences created using the two strategies: (1) 3 bp "scrambled motif prefix" 3 bp (2) original_sequence_start "scrambled motif prefix" original_sequence_end, through Fimo[23] with the two sets of TF motifs[24,25] ($P$ value $<10^{-4}$).
8. Choose the two "scrambled motifs" that result in the lowest number of motif hits indicated by the median rank across the two strategies, i.e., "non-motif sequences" that would be used on the array.

Approach 2—shuffle the motif:

In each chosen combination of region and motif, scramble the motif by shuffling its nucleotides.

*Library processing: replicates, association, barcode count, ratio.* Association: Reads from the association library were aligned to the reference set of sequences using bowtie2[73] with the *very-sensitive* preset parameters for maximal accuracy. A barcode was confidently assigned to a sequence if at least 3 unique UMIs supported that assignment and at least 80% of the UMIs associated with that barcode were aligned to the sequence. Barcodes that were not confidently assigned were considered ambiguous and discarded from downstream analyses. Overall, 7,004,354 barcodes were observed, of which 1,447,874 (20%) were confidently assigned, averaging 139.2 barcodes per sequence Supplementary Figs. 1 and 2). To make sure that our results are robust to the association thresholds, we repeated our analysis with a 99% threshold for the confident association, which resulted in highly consistent activity estimates (Pearson's correlation 0.97).

MPRA barcode counting: Reads from the MPRA libraries were processed against the set of confidently assigned barcodes, requiring a perfect match. Of the barcodes observed in the MPRA libraries, an average of 61.6% were confidently assigned, 37.4% were ambiguous (observed in the association library but were not confidently assigned), and 0.9% were unobserved in the association library (Supplementary Fig. 2). Only barcodes that appeared in at least two corresponding libraries (DNA and RNA libraries from the same time point and replicate) were included in downstream analyses, resulting in an average of 134.4 barcodes per sequence.

Quantification of induced transcription rate with MPRAnalyze: Quantification of induced transcriptional rates ("alpha" values) was performed using MPRAnalyze[22]. Briefly, MPRAnalyze fits two nested generalized linear models (GLMs): the first estimates the latent construct counts from the observed DNA counts, and the second estimates the latent rate of transcription from the latent construct estimates and observed RNA counts. The models are optimized using likelihood maximization, with a gamma likelihood for the DNA counts and a negative binomial likelihood for the RNA counts. MPRAnalyze includes library-size normalization factors, which were computed once using the entire dataset and then used across all analyses, including per-timepoint analyses, to maintain consistency. For quantification of alpha values, the full experimental design was included in the design matrix for the DNA model (~ timepoint + replicate + barcode), and an alpha value was extracted for each time point and replicate (RNA model: ~ timepoint + replicate).

Classification of active sequences with MPRAnalyze: Classification of active sequences was performed using the standard MPRAnalyze classification analysis, in which alpha values are mad-normalized (a median-based variant of z-normalization) and tested each value against the null distribution, estimated from the alpha values from the negative control scrambled sequences.

Comparative analyses with MPRAnalyze: The GLM structure of MPRAnalyze allows for a flexible framework to perform comparative analyses by using various design matrices for the different analyses (detailed below). Since the models are optimized using likelihood maximization, a likelihood ratio testing can be used for statistical significance and was used throughout all analyses in the manuscript. $P$ values were computed for each comparison and corrected within each analysis using Benjamini–Hochberg FDR correction[74].

For the per time point comparative analyses, each PERT and RAND sequence was compared with the corresponding WT sequence within each time point (DNA design: ~ replicate + barcode + sequence; Full RNA design: ~sequence; reduced RNA design: ~1). The resulting $P$ values were corrected jointly across all timepoints.

For temporal analyses, aimed at determining which sequences had temporal activity, we set the null behavior to be the temporal behavior exhibited by the scrambled sequences, by fitting a joint model to all SCRAM sequences and using the model coefficients as normalization factors for the comparative models (DNA design: ~ timepoint + replicate + barcode; Full RNA design: ~ timepoint; reduced RNA design: ~1).

For the comparative temporal analyses, we compared the temporal activity of each PERT or RAND sequence with the corresponding WT sequence, using an interaction term in the design (DNA design: ~timepoint + replicate + barcode; full RNA design: ~time * sequence; reduced RNA design: ~time). Note that the barcode covariate in the allele-comparative analyses (per-time point comparative analysis and temporal comparative analysis) is sequence-specific, so the barcode factor is confounded by the sequence variable.

Interaction analyses for multiple-perturbations using MPRAnalyze: The distribution of the joint perturbation design is as follows: for the same PWM joint perturbations (91): we have 19 that appear in one region, 6 in two regions, 4 in three regions, 2 in four regions, 2 in five regions, 2 in six regions, 1 in seven regions and 1 in eleven regions. For the different PWM joint perturbations (63): 5 appear

in two regions, 1 in three regions, 2 in four regions, 1 in eight regions, and 1 in nine regions. All the rest appear in one region.

We used MPRAnalyze to characterize the interactions between pairs of motifs by testing the hypothesis that corresponds to the billboard model of independent contribution, by assuming that the effects of each individual contribution is log-additive. We, therefore, included two binary covariates in the model: Pert1 indicated whether the observation comes from a sequence that contained the first Perturbation, Pert2 indicated whether it contained the second perturbation. In the full model, we then included an interaction term between these two covariates ($y \sim time + Pert1 * Pert2$), which we excluded from the reduced model ($y \sim time + Pert1 + Pert2$), so the effects will be independent. We then used a Likelihood Ratio Test to determine statistical significance, and the interaction coefficient was used as the interaction effect size.

Calculating RNA/DNA ratios: The calculation of RNA to DNA ratio is explained in detail in our previous work[20,59]. Briefly, to estimate the abundance of DNA or RNA per sequence and for each replicate (in order to compare replicates and time points), we use a simple averaging scheme:

13. $\text{D(R)NA per sequence} = \frac{10^6 * \sum_{i=1}^{\#BC} D(R)NA_i}{\#BC * sum(D(R)NA reads)}$ where D(R)NA$_i$ denotes the reads of a specific barcode i among the #BC barcodes that belong to the respective sequence.

To determine the RNA/DNA ratios per sequence and for each replicate we the sum of ratios:

14. $\left( \sum_{i=1}^{\#BC} \frac{RNA_i}{sum(RNA reads)} \Big/ \frac{DNA_i}{sum(DNA reads)} \right) / \#BC$

We added a pseudo count of 1 to the numerator and denominator to stabilize the signal from sequences with low numbers of reads. To combine replicates, we first divided the RNA/DNA ratios observed in each sample (time point/replicate) by the median ratio and then obtained the final RNA/DNA ratio by averaging the normalized values across replicates. We use the ratio calculation to compare the MPRA signal in this work to our previous work[20] (Supplementary Fig 4).

*Filtering sequences.* Filtering sequences per time point: We use MPRAnalyze to determine differential activity (explained in the previous section), for each perturbation method and each time point, comparing the following:

(PERT, WT), (RAND, WT), (PERT, RAND), (WT, SCRAM), and (PERT, SCRAM).

We use the following filters:

Filtering sequences per time point

1. We consider only sequences where WT (at each of the seven time points) **or** PERT have significantly different (MAD-score) regulatory activity than the null (SCRAM) (filter 3): length(FDR(WT, SCRAM) <0.05)==nof_TPs || FDR(PERT, SCRAM) < 0.05.
2. We only consider sequences where PERT has significantly different regulatory activity than its matching WT (filter 1): FDR(PERT, WT) < 0.05 1008, 1042, 998 out of (2082, 2086, 2114) sequences for perturbation methods 1, 2, and 3, respectively, pass these filters in at least one time point.

Filtering sequences across time

3. We consider only sequences where WT **or** PERT have significantly temporally different regulatory activity than the null (SCRAM) (filter 4). FDR(temporal(PERT,SCRAM)) <0.05 || FDR (temporal(WT,SCRAM)) <0.05).
4. and PERT has significantly temporally different regulatory activity than its matching WT (filter 2) FDR(temporal(PERT,WT)) < 0.05.

1189, 1224, 1354 out of (2082, 2086, 2114) sequences pass the temporal filtering for perturbation methods 1, 2, and 3, respectively.

We consider only sequences that are significant (pass all filtering steps per time point) in at least one time point and follow the temporal constraints, after filtering for duplicates, resulting in overall 747, 775, 749 sequences for perturbation methods 1, 2, and 3, respectively. Duplicates, i.e., sequences with motifs perturbed in the exact same locations (corresponding to different PWMs), were filtered, by picking the sequence with the lowest temporal FDR. FRSs are defined as sequences that pass all four filters **and** belong to the same main category (as described in the next section) in perturbation method 3 and either perturbation methods 1 or 2.

Filtering for pairs of motifs (double perturbation): IFRSs are defined as sequences with pairs of sites for which both single-site perturbations and the joint perturbation belong to the same main category and the double perturbation is functional (i.e passed all four filters as described above) in both perturbation approaches (perturbation method 3 and either of methods 1 or 2).

*Motif effect—main and sub-categories.* **Activators**—when this motif is perturbed in a region, the regulatory activity of PERT compared to WT is significantly reduced in at least one time point.

(i) Essential—this motif is essential for the regulatory activity of the region—i.e., scrambling this motif reduces the regulatory activity to null (SCRAM) or for all time points—the regulatory activity of PERT is similar to SCRAM.

FDR(temporal(PERT,SCRAM)) >0.05 || length(FDR_MAD(PERT,SCRAM) >0.05))==nof_TP

(ii) Contributing—this motif is contributing to the regulatory activity of the region—i.e., if we scramble this motif, the region is still regulatory active and its activity is different from null (SCRAM). If a motif is not essential, it is deemed contributing.

**Repressors**—when this motif is perturbed in a region, the regulatory activity of PERT compared to WT is significantly increased at at least one time point.

(iii) Silencing—this motif has a silencing effect on the regulatory activity of the region—i.e., the regulatory activity of the WT region is not temporally different from SCRAM or for all time points—the regulatory activity of WT is similar to SCRAM.

FDR(temporal(WT,SCRAM)) >0.05 || length(FDR_MAD(WT,SCRAM) >FDR_thresh))==nof_TP)

scrambling this motif increases the regulatory activity in at least one time point.

(iv) Inhibiting—this motif is reducing the regulatory activity of the region. If a motif is not silencing, it is deemed inhibiting.

*Activation dynamics analysis.* To examine the activation dynamics of activating FRSs, we examine activators that are active in all seven time points and fit a linear regression line to each FRS, modeling the absolute effect (WT–PERT) as a function of the WT activity level (delta ~ wt), using the *lm* function in R. the model parameters were then extracted and used for the extrapolation in Supplementary Fig. 9e).

*Statistics and reproducibility.* The 3 LentiMPRA replicates show high reproducibility (Supplementary Fig. 3). No statistical method was used to predetermine sample size. No data were excluded from the analyses. The experiments were not randomized. The investigators were not blinded to allocation during experiments and outcome assessment.

## Experimental procedures

*LentiMPRA library cloning and sequence-barcode association.* The lentiMPRA library construction was performed as previously described (Gordon et al.[28]). In brief, the array-synthesized oligo pool was amplified by 5-cycle PCR using forward primer (5BC-AG-f01, Supplementary Dataset 5) and reverse primer (5BC-AG-r01, Supplementary Dataset 5) that adds the minimal promoter (mP) and spacer sequences downstream of the sequence. The amplified fragments were purified with 1.8× AMPure XP (Beckman colter), and proceeded to second round 11-cycle PCR using the same forward primer (5BC-AG-f01) and reverse primer (5BC-AG-r02, Table Supplementary Dataset 5) to add 15-nt random sequence that serves as a barcode. The amplified fragments were then inserted into *Sbf*I/*Age*I site of the pLS-SceI vector (Addgene, 137725) using NEBuilder HiFi DNA Assembly mix (NEB), followed by transformation into 10beta competent cells (NEB, C3020) using the Gemini X2 machine (BTX). We note that there is not a typical polyA signal downstream of the WPRE in our lentiviral vector, as it was reported that an internal polyA signal can decrease virus titer[75]. Colonies were allowed to grow up overnight on Carbenicillin plates and midiprepped (Qiagen, 12945). We collected ~1 million colonies, so that on average 100 barcodes were associated with each sequence. To determine the sequences of the random barcodes and their association to each sequence, the sequence-mP-barcodes fragment was amplified from the plasmid library using primers that contain flowcell adapters (P7-pLSmP-ass-gfp and P5-pLSmP-ass-i#, Supplementary Dataset 5). The fragment was then sequenced with a NextSeq 150PE kit using custom primers (R1, pLSmP-ass-seq-R1; R2 (index read), pLSmP-ass-seq-ind1; R3, pLSmP-ass-seqR2, Supplementary Dataset 5) to obtain ~50 M total reads.

*Lentiviral infection and barcode sequencing.* Lentivirus was produced in twelve 15 cm dishes of 293T cells (CRL-3216, ATCC) using Lenti-Pac HIV expression packaging kit following the manufacture's protocol (GeneCopoeia, LT002). Lentivirus was filtered through a 0.45-µm PES filter system (Thermo Scientific, 165-0045) and concentrated by Lenti-X concentrator (Takara Bio, 631232). Titration of the lentiMPRA library was conducted on H1 human embryonic stem cells (WA-01, WiCell) as described previously[28]. Briefly, hESCs cells were plated at $1 \times 10^5$ cells/well in 24-well plates and incubated for 24 h. Serial volume (0, 4, 8, 16 µL) of the lentivirus was added with 8 µg/ml polybrene, to increase infection efficiency. The infected cells were cultured for three days and then washed with PBS three times. Genomic DNA was extracted using the Wizard SV genomic DNA purification kit (Promega). The multiplicity of infection (MOI) was measured as relative amount of viral DNA (WPRE region, WPRE.F and WPRE.R) over that of genomic DNA [intronic region of LIPC gene, LP34.F and LP34.R (Supplementary Dataset 5)] by qPCR using SsoFast EvaGreen Supermix (BioRad), according to the manufacturer's protocol. Lentiviral infection, DNA/RNA extraction, and barcodes sequencing were all performed as previously described[20].

Briefly, ~8 million cells (three 10-cm dishes) per time point were infected with the lentivirus library with a MOI of 5–8 along with 8 µg/mL polybrene (Sigma). Three independent replicate cultures were infected. To normalize technical bias of lentivirus preps, two of these replicates were infected with the same lentivirus batch, while the other replicate was infected with another lentivirus batch. The cells were incubated for 3 days with a daily change of the media. The infected cells were induced into neural

lineage using dual-Smad inhibition and harvested at 0 (right before differentiation), 3, 6, 12, 24, 48, and 72 h. DNA and RNA were purified using an AllPrep DNA/RNA mini kit (Qiagen). RNA was treated with Turbo DNase (Thermo Fisher Scientific) to remove contaminating DNA, and reverse-transcribed with SuperScript II (Invitrogen, 18064022) using barcodes-specific primer (P7-pLSmp-assUMI-gfp, Supplementary Dataset 5), which has a unique molecular identifier (UMI). Barcode DNA/cDNA from each replicate of each time point were amplified with 3-cycle PCR using specific primers (P7-pLSmp-assUMI-gfp and P5-pLSmP-5bc-i#, Supplementary Dataset 5) to add sample index and UMI. A second round of PCR was performed for 19 cycles using P5 and P7 primers (P5, P7, Supplementary Dataset 5). The fragments were purified and further sequenced with NextSeq 15PE with 10-cycle dual index reads, using custom primers (R1, pLSmP-ass-seq-ind1; R2 (index read1 for UMI), pLSmP-UMI-seq; R3, pLSmP-bc-seq; R4 (index read2 for sample index), pLSmP-5bc-seqR2, Supplementary Dataset 5).

**Reporting summary**. Further information on research design is available in the Nature Research Reporting Summary linked to this article.

## Data availability

The lentiMPRA data generated in this study have been deposited in the GEO database under accession code "GSE188264". All other relevant data supporting the key findings of this study are available within the article and its Supplementary Information files or from the corresponding author upon reasonable request. Source data are provided with this paper.

## Code availability

All code packages and pipelines are publicly available. The github links of the two pipelines that were used to analyze the data in this manuscript are "MPRAflow [https://github.com/shendurelab/MPRAflow]" and "MPRAnalyze [https://github.com/YosefLab/MPRAnalyze]". All custom code can be found on "zenodo [https://zenodo.org/record/5955738]" https://doi.org/10.5281/zenodo.5955738.

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

## Acknowledgements

This work was supported in part by the National Institute of Mental Health (NIMH) 1K99MH117393-01 and R00MH117393-03 (A.K.) and R01MH109907, U01MH116438 and R01MH125246 (N.A.), the National Human Genome Research Institute UM1HG009408 and UM1HG011966 (N.A.) and the Program for Breakthrough Bio-medical Research, which is partially funded by the Sandler Foundation (N.A.). The sequencing was carried by the DNA Technologies and Expression Analysis Core at the UC Davis Genome Center, supported by NIH Shared Instrumentation Grant 1S10OD010786-01.

## Author contributions

A.K., F.I., N.Y., and N.A. conceived and designed the study. A.K. and T.A. performed the computational analyses. F.I. performed the lentiMPRA experiments. C.D. assisted with experiments. A.K., T.A., F.I., N.Y., and N.A. interpreted the data. N.A. and N.Y. provided resources, and A.K., T.A., F.I., N.Y., and N.A. wrote the manuscript.

## Competing interests

N.A. is an equity holder of and a scientific advisor for Encoded Therapeutics, a gene regulation therapeutics company and is a co-founder of Regel Therapeutics. N.A. is also a co-inventor on related patent (Publication number WO/2018/148256) and patent (US Patent US2018017186) submitted by the University of California, San Francisco, that covers gene therapy for haploinsufficiency. N.Y. is an advisor for and/or has equity in Cellarity, Celsius Therapeutics, and Rheos Medicines. T.A. is an employee of Patch Biosciences. The remaining authors declare no competing interests.
