## [Peer Review File · Nature Communications]

Reviewers' Comments:

Reviewer #1:

Remarks to the Author:

Summary

In this study Kreimer et al. coupled TF motif mutagenesis in MPRA to assess functional variants of regulatory elements in high throughput. They systematically characterize the contribution of individual TF binding motifs and their interactions in the context of neural differentiation as a gradual and non-reversible model of cell fate determination.

Based on a previous study from the lab [Inoue et al, 2019], Kreimer et al. generate a lentiviral perturbation MPRA library of 591 regulatory elements with confirmed temporal activity during neuronal differentiation. With the help of the observed perturbation effects on transcription, TF motifs with temporal dynamics are newly classified into four categories of binding sites: essential, contributing, silencer, inhibitor. The authors claim that the direction of the observed effects is based on the DNA sequence, while the magnitude of effect depends on the cellular environment (for assessment of this statement see main point 4). With regards to the prevailing models of TF interaction the authors provide anecdotal evidence that regions with multiple essential functional regulatory sites support the "enhanceosome model", while regions with multiple contributing FRSs support a "billboard model" but without providing a systematic exploration of the results.

Overall Kreimer et al. present a carefully developed strategy for characterizing regulatory elements in a dynamic process, which results in a large-scale comprehensive data set over the course of neuronal differentiation. Several studies have utilized MPRA to characterize how TF motifs contribute to enhancer activity but have been focused on a limited number of cell types and on a specific time point. The publication of this study will provide a valuable resource for the research community and set an example for how MPRA can be used as a powerful and transferable tool. The MPRA displays a high reproducibility and includes well-designed controls, however the presentation and discussion of the data in the main figures and text make it difficult to assess the consistency of the results and the validity of the claims. This is because many results are shown and discussed based on very few examples that do not do the large-scale experiments justice and do not allow the reader to judge if these examples are representatives of consistent trends.

In sum, while the experiments provide a useful resource, the analysis is not exploiting the richness of the dataset and presents some claims in an anecdotal manner. We encourage the authors to address our major and minor comments on their manuscript. In addition, making the data and code available would be important given that the resource is an important aspect of this work.

Major comments

1 – The authors use and describe different filters to identify functional TF motifs which are included in their analysis (from 2,144 to 598). While the criteria applied seem very stringent, the description and reasoning for applying these filters require clarifications. Particularly the intuition for what is being selected for or against is missing here. This information needs to be included in order to enable the reader to assess the validity of these filters, their limitations, and to interpret the results.

- One example of a filtering step, which is hard to understand is the intuition of NOT requiring that $WT \neq \text{SCRAM}$ but instead that EITHER WT or PERT is $\neq \text{SCRAM}$. Do the authors want to account for repressive regions that can be de-repressed? In principle the authors already chose "regulatory regions that showed temporal activity" to start with and therefore WT should be $\neq \text{SCRAM}$ at some time point.

- It is also not clear whether the authors are removing constitutive enhancers due to the temporal dynamic filters? If so, why?

- Why do the authors include filters for temporal dynamics? If $WT/\text{PERT} \neq \text{SCRAM}$ and $\text{PERT} \neq \text{WT}$ at any time point, temporal dynamics should always be different. Figure 2, filter 2 left, shows an example where the dynamics is similar between the 3 types of sequences, with differences in the constant activity levels. This difference should be accounted for in filters 1 and 3, but this sequence should not pass filter 2 due to the similar dynamics. This example panel needs revision.

- In the text it is written that filter 3 requires $\text{PERT}/\text{WT} > \text{SCRAM}$ in all time points (line 246), but according to the methods and Fig 2A it should be a time point specific filter (i.e. "in all the time points" should be "in at least one time point"), correct?

Furthermore, it is not clear if the sequences/instances that did not pass the filters are because the regions had no activity or the motif instances are not important. This would be crucial information and the results and numbers of sequences/instances that pass each filter need to be shown and discussed in the main text.

Lastly, we wonder whether the motif instances that are not important are from the same set of TFs? Or are there TFs with some instances important and others not? This filtering strategy would benefit from support with ChIP data or ATAC-seq (e.g. from their previous Inoue et al, 2019) footprinting, to show that the selected functional sites are more occupied by the respective TFs.

2 – Based on the perturbation effects, the authors delineate 4 major categories of functional sites. However, the results are shown in a rather simplistic format (pie-charts, Fig. 2D), which is unfortunate given the comprehensiveness and quantitiveness of the data. The authors should illustrate better the results. One suggestion would be to include a summary of the importance of TF motifs across all regulatory sequences as a Supplementary Figure: For example, as a heatmap with 254 regions vs 147 TF motifs, colored by category (or no category if not present/important). This would allow the reader to see all the important motifs for a given sequence, or in which sequences there is a motif or motif combination important.

3 – Characterization of activating and repressive motif effects (Fig 3) – this section is too descriptive, highlighting only anecdotal examples, and requires more extensive analysis. The authors mention interesting cases of TFs, which they claim have consistent activating (SOX) or repressing (NEUROD2) effects, however they show only 3 regions as examples (Fig 3). In addition, it is not clear which TFs are activating or repressing and how consistent these effects truly are across instances for the same TF motif. The authors need to complement Fig 3 with panels illustrating these points. For example, the authors could add a plot that summarizes the number of instances of the different classes per TF motif (e.g. stacked barplot per TF). For demonstrating consistency of effects across instances, the authors could add to Fig 3 cumulative distribution plots (similar to Supp Fig 5) for 1 or 2 key TFs (e.g. Sox, NEUROD2) showing the WT, PERT and SCRAM activities for all mutated instances.

In addition, the number of TF motifs with mixed effects seems too high, but it is not clear if this is because a few instances are outliers or if the mix effect is robust (i.e. many activating and repressing instances). The more extensive analysis mentioned above should help address this point. The authors should also discuss what could explain how a TF motif can behave as an activator or a repressor in different regions, e.g. by the presence of partner motifs.

4 – One of the conclusions of this manuscript is that “the magnitude of the effects often varied over time (indicating that it depends on the cellular environment)” (abstract and lines 120,121). However, this is not supported by the data. As the authors acknowledge, the strength of the effect of mutations depends on the strength of the WT expression (line 318 and Supp Fig 8) - therefore it is not possible to interpret how the effects of the activator motifs change over time (Fig 4) without taking into account the WT activity of the respective sequences. The authors should correct these statements by mentioning this caveat or by toning down the claim in the abstract and lines 120,121 that otherwise are misleading.

The authors analyze the temporal motif effects (Fig 4) but there is no association with the expression dynamics of the respective TF. Their analysis of motif dynamics would gain in significance if the authors could integrate it with the expression of the predicted TFs per time point (RNA-seq data from their previous paper Inoue et al, 2019) to compare the timing of motif importance with the TF expression.

Additionally, the data is organized by hierarchical clustering. It would be interesting to see whether there are functional groups in each cluster. Could the observed clusters be interpreted as an early and a late cluster and are the TFs found here known to be involved at certain stages of differentiation?

Also, a scale is missing for Figure 4. And it does not become completely clear what “row normalized” means. Does that mean that a comparison between the TFs is not possible?

5 – The section on “characterization of genomic regions harboring sequence motifs” needs to be

substantially revised and expanded. The 2 paragraphs have essentially redundant information (page 20). We suggest that the authors combine this section with the motif pair section, describing which TFs in regions are essential or contributing, followed by how motif pairs interact, and only then combine the results and discuss them in light of the different enhancer sequence models.

6 – The section on the perturbation of motif pairs again only describes anecdotal examples (Fig 6) but misses a systematic assessment of the claim of different modes of interactions and a more global or systematic analysis of TF cooperativity:

The authors selected pairs of sites for which both single-site and joint perturbation belong to the same main category (lines 449-451). With this conservative selection and without showing the data before filtering it is difficult to assess the quality of the experiment and agreement between single-site and joint perturbation. They should more thoroughly discuss the selection in light of the observed initial results before filtering. In addition, the reason for such filtering (e.g. excluding motif pairs where single perturbations are from different categories) should be mentioned.

Additional technical details that should be included concern:

- The starting number of motif pairs before it was narrowed down to 22 pairs.
- Did the authors exclude motif pairs that overlap?
- In how many sequences does each pair occur on average?
- The authors should mention that with their selection they are not including cases where one motif instance is not important but has a super additive effect when mutated in combination.
- The authors should perform a more systematic analysis of TF cooperativity, otherwise it is not clear if the examples described are exceptional cases or represent a consistent cooperativity behavior between a TF pair. For example, the authors could compare for each pair the expected additive individual effect with the observed effect when mutating the pair, across all sequences.
- In Fig 5B, the classes of pairs are based on their motif individual classes, which is very confusing, since it does not take into account the effect of the joint perturbation.
- Wouldn't it be worth analyzing motifs, where the effect goes in both directions and the double mutation evens out? Compare the outcomes with the enhanceosome/billboard model expectations.

Minor comments

1 – The introduction is very long and needs to be revised. It contains too many technical details about MPRAs (page 3; page 5 99-100) and the explanations of previous MPRAs experiments/results are too detailed (Page 4,5). Instead, it is missing information on the hypotheses about enhancer sequences, TF motifs and the enhanceosome and billboard models. Including more information on the “models” tested here will help the reader to interpret the results later on and conclusions by the author. The paragraph about the summary of the work is also too extensive (Page 6).

2 - Histogram Fig 1b - Why are there bars lower than 20 in the x-axis if each motif should be perturbed in at least 20 regions?

3 - Colors used in Figure 2c/d are not matching; Legend: explain what method1/2/3 means.

4 - Figures: Is it possible to include error bars or confidence intervals for the time courses shown in Figures 3 and 6 (similar to the ones shown for SCRAM)? The lines are very thin and hard to distinguish in color but additionally error bars would make it a lot easier for the reader to estimate the variability of the observed effects. Additionally, the Figure legends need more details on what is graphed here.

Please also explain error bars in legend.

5 – The authors should discuss the 27 regions (21 + 6) that only contain repressor but not activator motifs (Fig 2D). This seems counterintuitive since the authors explicitly mention that none of the sequences repressed expression below negative control level. Did the authors miss/exclude activating motifs? For example, were activator motifs filtered out because they were not functional or do they contain activator motifs that were not selected for mutagenesis? Or do these sequences truly not contain activator motifs? A clarification is warranted here.

6 - The claim made in the text that SOX1/2 seems to be a pioneering factor and therefore a disruption of the motif fully disrupts activity and falls in the essential category seems a bit vague.

It is unclear why this example was chosen and how this relates to the other transcription factors. How many of them have been described as pioneering factors before?

7 - Legends of the Supplementary Figures are largely insufficient. Supp Fig 9 also poor quality: cannot be read and the colour legend is not clear

8 - Split section "lentiMPRA perturbation" of results - it would help adding a new section before line 238 with the "identification of functional TF motifs"

9 - Typos:

Line 69 - add "s" to MPRA"s"

Line 283 - it should be 72 "that had repressing effects"

Line 313 - "12 silencers and 60 inhibitors" don't match the numbers in Fig 2D

Line 342 - should be Fig 3a

Line 394 - should be Supp Fig 9

Line 427 - add percentage (as in line 429)

Line 451 - clarify what "main categories" you mean

Line 760 - correct "scrambled"

Line 827 - correct "RNAD"

Reviewer #2:

Remarks to the Author:

Review of Kreimer et al:

This paper addresses an important question about the basic biology of how enhancers function across different cell types as cell differentiate from iPSCs to NPCs. It leverages lentiMPRA technology, elegantly applied to the differentiating system, which enables the same library of elements to be assessed in different cell types. It is quite clearly written and the authors claim to find several general principles of enhancers from their study of these 591 genomic elements and perturbation of 255 motifs within these, including about 20 double perturbations. They anchor it in the system, time points, and data from their 2019 Cell Stem Cell paper, allowing them good reference back to the cellular context.

Major claims of the paper: 1) There are 4 categories of functional elements. 2) Activity direction is determined by the element itself, but magnitude is determined by the cellular context. 3) Fine tuning of expression levels is mediated by adjacent elements activating and repressing. 4) there sometimes is cooperation between motifs, and both the enhanceosome and billboard model fit the data in some cases.

There are a few spots where I had some statistical concerns that need to be addressed with re-analysis or writing, perhaps one experiment to better query repressive/silencing sequences. Also, the presentation of the data could be clarified in several of the figures, supplemental figures, and especially legends and I have made some suggestions. If the statistical concerns are mitigated by the added analyses and/or experiment, overall I think the paper could be suitable with a substantial revision.

Review:

Substantial concerns:

Line 780: I am concerned that allowing 20% of the barcodes to come from a different sequence might be attenuating some of their results. What happens if you required a 99% of the reads for a barcode to align to a single genomic element? I imagine you would end up with less FRS regions for subsequent analysis, and so a less comprehensive dataset. But, if you do this, are the major conclusions of the paper still the same? If so, then it might be fine to present the 80% data but discuss in methods that the findings are robust to this parameter.

Line 298, and generally their ability to assess silencing sequences. Is it possible the minP is too weak to enable seeing repressive sequences? After all, you need a fair amount of expression

present to repress it. Or would very repressive sequences be eliminated from the analysis by filtering (i.e., no RNA counts? – line 790)? Or the ARE's in the construct preventing them from having their effect? Certainly they could try adjusting their filtering to account for the 'no RNA count' filter preventing discovery of repressive elements and see if more are detected. Otherwise, perhaps these points just warrant discussion and a different library design is needed to better understand repression (i.e. no AREs and a higher activity minP).

Figure 3 (and similar panels on later figures): Why are there no error bars on the individual element perturbations? There are 3 biological replicates as well as multiple barcodes for each? And multiple different ways of perturbing each. There should be a reasonable way to represent some experimental error on these figures.

When motifs really overlap a lot (4d, 6b) how do you interpret perturbations in them? Is it possible to perturb one motif and not the other and did they design to make sure this was the case? How does this change the interpretation relative to the enhanceosome vs billboard models? For example, are the overlapping ones more likely to fall into the enhanceosome model, while more distal ones act more like billboards?

For their discussion section, if neither the enhanceosome model nor the billboard model fit the data, can the authors propose an alternative that does fit all the data?

Lines 211-214, is there a supplemental figure that goes with this data? Statistical tests to support the assertions?

Minor concerns:

Figure 6 sounds like they are thinking about testing for main effects of each mutation, and then testing for an interaction term. Are they? It seems that would be the most straightforward way to be confident when there is an interaction. Is their nested model compatible with including such an interaction term? Then they could confidently state when the interaction between the two perturbations is significant.

I find the section around line 321/ Supplementary figure 8 hard to follow. For example, the terms a and b on line 319 are not defined. Is everything here in log2FC rather than just FC? In addition, I am confused about the discrepancy between 8a-c (which show a linear relationship) and 8d-e, which argue for a nonlinear relationship using the same data? Are they just showing examples at the low end of expression a-c? Or extrapolating to what would happen beyond expression levels that they measure? What is the major conclusion of this section?

I assume there is a poly A signal in their reporter too downstream of the WPRE?

Supplemental figures should have legends to make them easier to interpret. I was often left guessing what some panels or axis labels meant. Fonts on figures often quite small. The authors should be applauded for including thorough supplemental tables that others might use for reanalysis in the future. However, the legends for these should tell potential end-users what the different sheets are, and what the different columns are and how to interpret them (e.g., in the case of coding variables, or the second row of some of the spreadsheets that appears to be some kind of numerical code?).

It's a little unclear (e.g., around 295) whether each sequence gets compared to its own scrambled control? Or all scrambled controls?

Very minor/grammatical:

I am trying to figure out what a perturbation MPRA is (line 131). Is it mutating the motifs? Or assaying across time? Either way, I agree it is powerful, but testing expression with and without motif disruptions is what MPRA have been used for a long time. I'm not sure I'd emphasize a new name too much for an old thing. This is not a big deal though.

Sentence on line 87 might better be broken into 2 sentences.

Line 117 makes it sound like they are evaluating the endogenous target gene. They might want to change this to "reporter gene."

The four categories they mention in their abstract seem more pre-defined once I read the paper, rather than discovered from the data as the abstract might be read to imply. They should perhaps rephrase this a bit in the abstract.

Figure 3 and similar panels in 4: The key took a minute to parse, as it includes both the name of the perturbed element, some unique coded identifier, and the interpretation of the effect (Contributing, Essential, ect). The authors might want to think about how to present this in a way that separates these things a bit and makes it easier to understand the data quickly at a glance. Also, the effect of Nanog was so small I almost didn't notice it in 3b. Fonts got to be quite small as well, and making the lines a bit thicker in their line weight would make it easier to distinguish the colors, which I struggled with. Generally in the paper the data presentation could be polished up a bit.

There are two 6d's in the legend.

272: extra space

374: refs formatted strangely.

There is no supplemental primer table?

Reviewer #3:

Remarks to the Author:

Kreimer et al have taken the next step in the productive use of MPRA technology by providing a robust high-throughput framework for perturbing potential regulatory sequences, specifically querying transcription factor (TF) binding sites; by instantiating a perturbation element, a GFP reporter, and a set of filters for the assignment of higher-probability functional regulatory elements in the genome.

The authors explore this approach along 7 timepoints in the progression of human ES cells to neural progenitors; a developmental pathway that has been demonstrated to be a highly fertile realm for transcriptional regulation. This important and valuable step is validated in two ways: the binding sites occur on genes known to be involved in early neural differentiation and reporter readouts are altered along the timecourse of development. This allows the authors to provide insights into the "regulatory architecture" of the journey of stem cells to neural progenitors.

Their approach is easy to follow and allowed for efficient filtering to focus on a more limited set of functionally regulatory sequences (FRS). The classification system in which the FRS were categorized is supported by previous studies and their results provide nuance for the current models of gene expression regulation by distal sequences (ie, the "enhancesome" and "billboard" models). Two strengths of this work are that: 1) it provides a framework for investigating the relationship of TFs within the same regulatory region, which will likely help to understand large and highly complex regulatory regions that are perturbed in developmental disorders, and 2) the procedures could have significant uptake in any lab where such studies are being contemplated. They have also developed previously published tools, ie MPRAflow and MPRAalyze, to assist in utilizing this system.

Three issues with the data presented should be addressed. 1). The term "temporal invariance" (line287), is applied to what seems to refer to whether an element is an activator or a repressor at times examined. But the term literally suggests that time does not play a role; that is, expression is invariant over time which is clearly not the case in the examples shown or from the many investigations that show temporal dynamicism vs invariance. 2). A second point that seems

counter intuitive is that the repressive elements are found to be no lower than baseline/control; one would expect a lower-than baseline impact for any inhibitory elements and this causes concern. 3). Since the FRS perturbations are randomly integrated in this system, it's hard to believe that the impact on enhancer activity will occur in the same manner in its native chromatin environment and here functional validation beyond decreased expression would be very helpful (see below).

A potential strength of the report would have been to take a known and a novel regulatory element to the stage of functional validation. This step would have provided compelling and demonstrative value to this approach. If this had been demonstrated, then this contribution would have taken the lead in the field; both for detailing the higher throughput analysis with perturbations as detailed, but also with the follow through of a functional validation step which is something that only currently stands as a promissory note; while an outright deliverable is what is desired. As this reviewer sees the field, there has been an evolution of the MPRA technology over the last decade-plus that is in need of coupling with a biological context, and this paper does not take this important step.

The types of validation yielding biological context that one would really like to see would be to include a phenotypic analysis on the impact of these perturbations on neuron differentiation. This would include showing that the TF is actually bound to a given FRS with some temporal specificity that points to a perturbed developmental process. This study should provide some type of evaluation of whether the effect that is seen on the reporter gene/transcribed barcode, actually translates to differences in gene target regulation and phenotypic outcomes.

Thus, what the authors did not do, which would have created a more enthusiastic uptake from this reviewer, was to address questions such as: What role do FRSs play during neuronal differentiation? Which type of FRS may be responsible for driving cell identity or hESC maintenance and proliferation? Are the TFs found in interacting FRS novel or are they known regulators? For what reason would a FRS need "fine tuning" using activators and silencers in the context of neural development?

It is my thought that a Nature Comms paper should provide some reasonable answers to these queries. Currently, this paper is an elegant piece of work and should easily find its place in a high profile techniques-oriented journal.

We thank the editor and reviewers for their incredibly helpful comments. We revised our manuscript, figures, and supplementary material to reflect these changes and new results (all changes in these files are in blue font so that can be easily observed). These are also described in our point-by-point response to reviewers below. Reviewer comments are in black text and our responses are in blue text.

REVIEWER COMMENTS

Reviewer #1:

Summary

In this study Kreimer et al. coupled TF motif mutagenesis in MPRA to assess functional variants of regulatory elements in high throughput. They systematically characterize the contribution of individual TF binding motifs and their interactions in the context of neural differentiation as a gradual and non-reversible model of cell fate determination. Based on a previous study from the lab [Inoue et al, 2019], Kreimer et al. generate a lentiviral perturbation MPRA library of 591 regulatory elements with confirmed temporal activity during neuronal differentiation. With the help of the observed perturbation effects on transcription, TF motifs with temporal dynamics are newly classified into four categories of binding sites: essential, contributing, silencer, inhibitor. The authors claim that the direction of the observed effects is based on the DNA sequence, while the magnitude of effect depends on the cellular environment (for assessment of this statement see main point 4). With regards to the prevailing models of TF interaction the authors provide anecdotal evidence that regions with multiple essential functional regulatory sites support the “enhanceosome model”, while regions with multiple contributing FRSs support a “billboard model” but without providing a systematic exploration of the results.

Overall Kreimer et al. present a carefully developed strategy for characterizing regulatory elements in a dynamic process, which results in a large-scale comprehensive data set over the course of neuronal differentiation. Several studies have utilized MPRA to characterize how TF motifs contribute to enhancer activity but have been focused on a limited number of cell types and on a specific time point. The publication of this study will provide a valuable resource for the research community and set an example for how MPRA can be used as a powerful and transferable tool. The MPRA displays a high reproducibility and includes well-designed controls, however the presentation and discussion of the data in the main figures and text make it difficult to assess the consistency of the results and the validity of the claims. This is because many results are shown and discussed based on very few examples that do not do the large-scale experiments justice and do not allow the reader to judge if these examples are representatives of consistent trends.

In sum, while the experiments provide a useful resource, the analysis is not exploiting the richness of the dataset and presents some claims in an anecdotal manner. We encourage the authors to address our major and minor comments on their manuscript. In addition, making the

data and code available would be important given that the resource is an important aspect of this work.

Major comments

1 – The authors use and describe different filters to identify functional TF motifs which are included in their analysis (from 2,144 to 598). While the criteria applied seem very stringent, the description and reasoning for applying these filters require clarifications. Particularly the intuition for what is being selected for or against is missing here. This information needs to be included in order to enable the reader to assess the validity of these filters, their limitations, and to interpret the results.

We thank the reviewer for this comment and completely agree that the rationale behind the filters should be better explained. To address this, we added the following clarifications in the revised manuscript:

- One example of a filtering step, which is hard to understand is the intuition of NOT requiring that $WT \neq SCRAM$ but instead that EITHER WT or PERT is $\neq SCRAM$. Do the authors want to account for repressive regions that can be de-repressed? In principle the authors already chose “regulatory regions that showed temporal activity” to start with and therefore WT should be $\neq SCRAM$ at some time point.

The reviewer is correct. The reason why we require EITHER WT or PERT is $\neq SCRAM$ is to account for both repressive and active sequences. We now added this explanation in the main text. Additionally, we chose “regulatory regions that showed temporal activity” in our previous publication, but there are a few cases where such (borderline) sequences come as not temporarily active in this assay and those account for the few silencer examples we found.

- It is also not clear whether the authors are removing constitutive enhancers due to the temporal dynamic filters? If so, why?

We are not removing constitutive enhancers as long as their temporal activity is significantly different from the WT (e.g. constant higher activity will be called as significant per the MPRAnalyze model). This is now clarified in the main text.

- Why do the authors include filters for temporal dynamics? If $WT/PERT \neq SCRAM$ and $PERT \neq WT$ at any time point, temporal dynamics should always be different. Figure 2, filter 2 left, shows an example where the dynamics is similar between the 3 types of sequences, with differences in the constant activity levels. This difference should be accounted for in filters 1 and 3, but this sequence should not pass filter 2 due to the similar dynamics. This example panel needs revision.

As mentioned in the previous response, our analysis is not removing constitutive enhancers as long as their temporal activity is significantly different from the WT. i.e. a constant difference would be recorded as significant, so this sequence should pass filter 2.

- In the text it is written that filter 3 requires $PERT/WT > SCRAM$ in all time points (line 246), but according to the methods and Fig 2A it should be a time point specific filter (i.e. “in all the time points” should be “in at least one time point”), correct?

This is now clarified in the main text: “Either the PERT (in at least one time point) or WT (in all the time points) sequences are significantly more active than the SCRAM negative controls.”

Furthermore, it is not clear if the sequences/instances that did not pass the filters are because the regions had no activity or the motif instances are not important. This would be crucial information and the results and numbers of sequences/instances that pass each filter need to be shown and discussed in the main text.

To address this, we added a new **Supplementary Fig. 7** indicating for each tested sequence in each perturbation method, which filters it passed. We now point to the figure from the main text along with the following text: “Across the 3 perturbation methods, we observe that most of the sequences pass all 4 filters and less than 10% of the sequences pass no filter, indicating that our experimental design mainly consists of functional regulatory sequences across these time points of neural induction (**Supplementary Fig. 7**).”

Lastly, we wonder whether the motif instances that are not important are from the same set of TFs? Or are there TFs with some instances important and others not? This filtering strategy would benefit from support with ChIP data or ATAC-seq (e.g. from their previous Inoue et al, 2019) footprinting, to show that the selected functional sites are more occupied by the respective TFs.

We thank the reviewer for this useful comment. We added to the revised version the following analysis in the main text “Comparison analysis of temporal properties from our previous work (Inoue et al. 2019) confirmed that the signal of H3K27ac, ATAC-seq, MPRA and mRNA of both the closest gene and the motif’s associated TF were significantly lower (all time points

combined, Wilcoxon p -value $< 10^{-10}$) in filtered vs. chosen sequences, supporting our filtering approach.“ Finally, most of the motifs associated with the same TF are found in both chosen and filtered sequences, supporting our conclusion that the motif effect is dependent both on the motif and the surrounding sequence.

2 – Based on the perturbation effects, the authors delineate 4 major categories of functional sites. However, the results are shown in a rather simplistic format (pie-charts, Fig. 2D), which is unfortunate given the comprehensiveness and quantitiveness of the data. The authors should illustrate better the results. One suggestion would be to include a summary of the importance of TF motifs across all regulatory sequences as a Supplementary Figure: For example, as a heatmap with 254 regions vs 147 TF motifs, colored by category (or no category if not present/important). This would allow the reader to see all the important motifs for a given sequence, or in which sequences there is a motif or motif combination important.

We thank the reviewer for this suggestion, and created a heatmap of motifs and regions (**Supplementary Fig. 10**). The heatmap is sparse, but shows some patterns in the data: e.g groups of similar regions contain the same motifs; related motifs that tend to appear in the same regions; and importantly - that motifs have different effects in different regions. A more detailed visualization of this is available in **Supplementary Fig. 11**, which shows the distribution of categories for each motif and TF. We also included the following text: “When examining the distribution of motif effects across regions (**Supplementary Fig. 10**), we observe that related motifs tend to appear in the same regions and importantly - that motifs have different, in many times opposing, effects in different regions. This is also supported by a per motif visualization showing the distribution of categories per motif (**Supplementary Fig. 11**). Additionally, there are groups of similar regions that contain the same motifs (**Supplementary Fig. 10**).“

3 – Characterization of activating and repressive motif effects (Fig 3) – this section is too descriptive, highlighting only anecdotal examples, and requires more extensive analysis. The authors mention interesting cases of TFs, which they claim have consistent activating (SOX) or repressing (NEUROD2) effects, however they show only 3 regions as examples (Fig 3). In addition, it is not clear which TFs are activating or repressing and how consistent these effects truly are across instances for the same TF motif. The authors need to complement Fig 3 with panels illustrating these points. For example, the authors could add a plot that summarizes the number of instances of the different classes per TF motif (e.g. stacked barplot per TF). For demonstrating consistency of effects across instances, the authors could add to Fig 3 cumulative distribution plots (similar to Supp Fig 5) for 1 or 2 key TFs (e.g. Sox, NEUROD2) showing the WT, PERT and SCRAM activities for all mutated instances.

To address this comment, we generated a novel figure showing a stacked barplot for each motif and aggregated by all motifs of a specific TF. This figure is showing the distribution of motif effects across the 4 categories (**Supplementary Fig. 11**). The only factor for which we see a consistent categorization is SOX1, which is almost exclusively activating and is enriched for essential function, which can be seen in this visualization. For most other motifs the effects are

mixed. NEURO2D only appears in one Functional Regulatory Sequence (FRS) that passes the filters, and is indeed silencing in this example, but we do not claim that it is consistently repressive. We have clarified all of this in the following text:

“ Within the activating FRSs, we observed that motifs associated with the SRY-Box Transcription Factor SOX1 are **the only motifs that are enriched** in the set of *essential* FRSs (i.e. over-representation that is unlikely to occur by chance; hypergeometric test, FDR<0.05; **Supplementary Fig. 11**).”

“For most of the motifs the effects are mixed (**Supplementary Fig. 11**). These results indicate that enhancer activity is influenced both by the motif sequence and the surrounding sequence of the region harboring the motif (**Supplementary Figs. 10-11**).”

In addition, the number of TF motifs with mixed effects seems too high, but it is not clear if this is because a few instances are outliers or if the mix effect is robust (i.e. many activating and repressing instances). The more extensive analysis mentioned above should help address this point. The authors should also discuss what could explain how a TF motif can behave as an activator or a repressor in different regions, e.g. by the presence of partner motifs.

We thank the reviewer for the comment, and point to **Supplementary Fig. 11** mentioned above, which shows that the mixed effects are fairly common in our data. There are examples of factors that are exclusively activating, but we found that repressive motifs usually have mixed effects if they appear in sufficiently many regions, indicating the repressive activity is generally context-dependent. We mention this in the text, however, discussing the specific molecular mechanisms that explain the abundance of mixed-effect functions is currently quite speculative, since we can't tell if it is a result of partner motifs, factor modifications, or other cellular interactions.

4 – One of the conclusions of this manuscript is that “the magnitude of the effects often varied over time (indicating that it depends on the cellular environment)” (abstract and lines 120,121). However, this is not supported by the data. As the authors acknowledge, the strength of the effect of mutations depends on the strength of the WT expression (line 318 and Supp Fig 8) - therefore it is not possible to interpret how the effects of the activator motifs change over time (Fig 4) without taking into account the WT activity of the respective sequences. The authors should correct these statements by mentioning this caveat or by toning down the claim in the abstract and lines 120,121 that otherwise are misleading.

We thank the reviewer for pointing this out. We do observe that the magnitude of the effects varies over time, and that it also scales linearly with the WT transcription levels (**Supplementary Fig 9**). We also point out that the WT transcriptional levels themselves depend on the cellular environment. The WT levels depend on the cellular environment and change over time, and the effect of a perturbation scales linearly with it, and therefore also depends on the cellular environment. We have clarified this in the text by adding the following: “*These results suggest that while the relationship between WT activity and the effect of perturbation is linear, both depend on the sequence content and the specific cellular context in which it is being assayed.*”

The authors analyze the temporal motif effects (Fig 4) but there is no association with the expression dynamics of the respective TF. Their analysis of motif dynamics would gain in significance if the authors could integrate it with the expression of the predicted TFs per time point (RNA-seq data from their previous paper Inoue et al, 2019) to compare the timing of motif importance with the TF expression.

We tested this correlation and haven't seen conclusive results. This is not surprising, since our analysis is motif-based, and since similar sequence motifs are not independent, it is likely that the annotation of the FRSs suffers from misclassification of the binding factor. Additionally, even if the exact factor was known, it is not established in current literature that the magnitude of TF gene expression is directly correlated with its regulatory effect, so a strong correlation is not necessarily expected. We have added the following to the text to address this in the discussion: “we used RNA-seq data from (Inoue et al. 2019) to compare the timing of motif importance with the respective TF expression. Testing this correlation did not show conclusive results. We speculate that this is due to the nature of our analysis which is motif-based, and since similar sequence motifs are not independent, it is likely that the annotation of the FRSs suffers from misclassification of the binding factor. Additionally, even if the exact factor was known, it is not established in current literature that the magnitude of TF gene expression is directly correlated with its regulatory effect, so a strong correlation is not necessarily expected. “

Additionally, the data is organized by hierarchical clustering. It would be interesting to see whether there are functional groups in each cluster. Could the observed clusters be interpreted as an early and a late cluster and are the TFs found here known to be involved at certain stages of differentiation?

Also, a scale is missing for Figure 4. And it does not become completely clear what “row normalized” means. Does that mean that a comparison between the TFs is not possible?

We thank the reviewer for their comment, and now add the following in the text: “Enrichment analysis of the TFs (both activating and repressing) in the early cluster (**Fig. 4a-b**) indicated their involvement in processes related to cell differentiation, cell fate commitment and regulation of development for the top 10 categories, whereas enrichment of late response TFs (**Fig. 4a-b**) indicated, more specifically, categories related to neurogenesis and nervous system development (Subramanian et al. 2005). These results support the functionality of these clusters in earlier and later stages of neural differentiation.”

Additionally, we added a scale to Figure 4. As the reviewer noted, ‘row normalized’ indicates that the color scale in each row (TF) represents the comparative strength of the signal for that TF and is not comparable across the rows.

5 – The section on “characterization of genomic regions harboring sequence motifs” needs to be substantially revised and expanded. The 2 paragraphs have essentially redundant information (page 20). We suggest that the authors combine this section with the motif pair section, describing which TFs in regions are essential or contributing, followed by how motif pairs

interact, and only then combine the results and discuss them in light of the different enhancer sequence models.

The suggested edits are now incorporated in the main text.

6 – The section on the perturbation of motif pairs again only describes anecdotal examples (Fig 6) but misses a systematic assessment of the claim of different modes of interactions and a more global or systematic analysis of TF cooperativity:

The authors selected pairs of sites for which both single-site and joint perturbation belong to the same main category (lines 449-451). With this conservative selection and without showing the data before filtering it is difficult to assess the quality of the experiment and agreement between single-site and joint perturbation. They should more thoroughly discuss the selection in light of the observed initial results before filtering. In addition, the reason for such filtering (e.g. excluding motif pairs where single perturbations are from different categories) should be mentioned.

We thank the reviewer for the opportunity to clarify this section. To address all comments from the reviewers, we completely revised the ‘**Perturbation of motif pairs identifies different modes of motif interaction**’ section by implementing a statistical model to test for pair interaction and accordingly revised all the relevant parts under the Discussion and Methods sections. We additionally revised Figure 5 (replacing previous Figs. 5-6). This is now revised in the main text and all other relevant parts in the manuscript:

“We considered the FRSs to have independent effects (following the billboard model) if the effects were log-additive: perturbing both sites was equivalent to multiplying the effects of perturbing each site separately. We used MPRAnalyze²² to test this hypothesis for each assayed pair in each perturbation method, by including an interaction term in the model that captures the effect of perturbing both sites while accounting for the effect of perturbing both sites individually (**Methods**). We considered pairs to have significant interaction if the size of the interaction term was larger than 0.5 and the test was statistically significant (BH-corrected $p < 0.05$). We then defined interaction as “consistent” if the pair were labeled the same (either significant or non-significant) in perturbation methods 3 and either 1 or 2, and removed inconsistent pairs from the analysis. We then also removed pairs in which none of the perturbations are functional, by requiring that at least one of the single perturbations pass the filtering scheme we described above. Finally, to make interpreting the results easier, we excluded pairs in which the assayed sites overlap since overlapping sites cannot be conclusively independent. Overall, out of 149 examined pairs, 24 pairs remained, of which 13 were log-additive, consistent with a billboard model of cooperation, and 11 had significant non-additive interactions (**Fig. 5b**). While the small number of functional pairs in our results does not allow for extensive or systemic analyses, we do find anecdotal evidence of different cooperation models operating in different regions.”

Additional technical details that should be included concern:

- The starting number of motif pairs before it was narrowed down to 22 pairs.

The design includes 149 motif pairs. In the revised analysis, after removing overlapping motifs and pairs with inconsistent patterns of interaction, we remain with 24 consistent pairs to include in the analysis. This has been clarified in the revised text: *Overall, out of 149 examined pairs, 24 pairs remained, of which 13 were log-additive, consistent with a billboard model of cooperation, and 11 had significant non-additive interactions.*

- Did the authors exclude motif pairs that overlap?

While we originally included overlapping pairs in the design since they represent real biological phenomena, we found these challenging to interpret since independence cannot be assumed, and therefore excluded them from the analysis.

- In how many sequences does each pair occur on average?

In the final set of pairs included in our analysis (total 24 pairs), each motif pair occurs in a single region, with the exception of two same-pwm pairs that appear in two sequences (RFX5_disc3, BHLHE40_disc2).

- The authors should mention that with their selection they are not including cases where one motif instance is not important but has a super additive effect when mutated in combination.

We thank the reviewer for this comment. In the revised analysis we now include pairs in which one of the motifs is insignificant in isolation. One such example is included in the revised Figure 5g and discussed in the text: *In chr16:51185391-51185562 (hg19), upstream of the promoter of neurogenesis regulator SALL1, we find two binding sites of TRIM28. When perturbed individually, one site has no effect, while the other has a mild dampening effect. However, when both are perturbed the effect is a significant decrease in activity. This potentially demonstrates a redundancy mechanism, whereby either binding site is sufficient for the WT activity, and both need to be perturbed in order to disrupt it (Fig. 5g)*

- The authors should perform a more systematic analysis of TF cooperativity, otherwise it is not clear if the examples described are exceptional cases or represent a consistent cooperativity behavior between a TF pair. For example, the authors could compare for each pair the expected additive individual effect with the observed effect when mutating the pair, across all sequences. We thank the reviewer for this suggestion and have indeed revised the analysis of motif pairs to include a more extensive analysis of cooperativity, including modeling the expected vs. observed effect of the double-perturbation. The small number of consistent interpretable pairs did not allow for extensive characterizations, but the revised analysis does offer more quantitative and rigorous analysis of cooperation patterns.

- In Fig 5B, the classes of pairs are based on their motif individual classes, which is very confusing, since it does not take into account the effect of the joint perturbation.

We agree with the reviewer. The revised analysis no longer uses the individual effects to categorize the double perturbation.

- Wouldn't it be worth analyzing motifs, where the effect goes in both directions and the double mutation evens out? Compare the outcomes with the enhanceosome/billboard model expectations.

We thank the reviewer for the suggestion. In the motif pairs included in our analysis we didn't find an example of the effect cancelling out (i.e the double perturbation being equivalent to the WT, despite both single perturbations having an effect). However, we did find a billboard-consistent example of two motifs that individually had an opposite effect (one activating, one dampening), and together had a mitigated effect. From the revised text: *we find that additive contribution can also apply to cooperating activators and dampeners, as in chr4:152405951-152406122 (hg19), an intronic region in the FAM106A1 gene body, which contains an FRS with a SOX1 motif that has a contributing effect and an FRS with a ZIC2 motif that has an inhibiting effect. Perturbing both sites results in an additive effect: transcription levels that are lower than WT, but higher than those obtained when perturbing the SOX1 motif alone (Fig. 5d).*

Minor comments

1 – The introduction is very long and needs to be revised. It contains too many technical details about MPRAs (page 3; page 5 99-100) and the explanations of previous MPRA experiments/results are too detailed (Page 4,5). Instead, it is missing information on the hypotheses about enhancer sequences, TF motifs and the enhanceosome and billboard models.

Including more information on the “models” tested here will help the reader to interpret the results later on and conclusions by the author. The paragraph about the summary of the work is also too extensive (Page 6).

The Introduction is now shortened and includes information about the models.

2 - Histogram Fig 1b - Why are there bars lower than 20 in the x-axis if each motif should be perturbed in at least 20 regions?

This is because the graph contains 1,547 temporal regions we detected in our previous paper and Fig. 1b only shows the 591 selected regions and the motifs in them.

3 - Colors used in Figure 2c/d are not matching; Legend: explain what method1/2/3 means.

The colors have been fixed to match. Method 1/2/3 refers to the perturbation method as explained prior in the main text.

4 - Figures: Is it possible to include error bars or confidence intervals for the time courses shown in Figures 3 and 6 (similar to the ones shown for SCRAM)? The lines are very thin and hard to distinguish in color but additionally error bars would make it a lot easier for the reader to estimate the variability of the observed effects. Additionally, the Figure legends need more details on what is graphed here.

Please also explain error bars in legend.

All figures in the main text and supplementary have been revised to include error bars, using the three replicates to estimate the standard deviation. The legend is now revised for all appropriate figures.

5 – The authors should discuss the 27 regions (21 + 6) that only contain repressor but not activator motifs (Fig 2D). This seems counterintuitive since the authors explicitly mention that none of the sequences repressed expression below negative control level. Did the authors miss/exclude activating motifs? For example, were activator motifs filtered out because they were not functional or do they contain activator motifs that were not selected for mutagenesis? Or do these sequences truly not contain activator motifs? A clarification is warranted here.

We thank the reviewer for this comment and agree that clarification is needed. We assume that these regions contain additional activating FRSs that were not included in our design. To clarify this in the text, we included the following: “Notably, these FRSs are not a comprehensive list of all functional sites in the selected regions. For instance, we found several regions in which only repressive FRSs were identified (**Fig. 2d**). Since repressive FRSs only reduce the overall activating function of the region, repressive-only regions must contain additional unknown activating FRSs that were not included in our design.”

6 - The claim made in the text that SOX1/2 seems to be a pioneering factor and therefore a disruption of the motif fully disrupts activity and falls in the essential category seems a bit vague. It is unclear why this example was chosen and how this relates to the other transcription factors. How many of them have been described as pioneering factors before?

This is now clarified in the main text: SRY-Box Transcription Factor SOX1 are the only motifs that are enriched in the set of *essential* FRSs (i.e over-representation that is unlikely to occur by chance; hypergeometric test, FDR<0.05).

7 - Legends of the Supplementary Figures are largely insufficient. Supp Fig 9 also poor quality: cannot be read and the colour legend is not clear

All the supplementary data has been significantly revised to have more information and be of better quality.

8 - Split section “lentiMPRA perturbation” of results - it would help adding a new section before line 238 with the “identification of functional TF motifs”

We revised the results section to have these as two separate sections.

9 - Typos:

Line 69 - add “s” to MPRA”s” - fixed

Line 283 - it should be 72 “that had repressing effects”

It is actually 70, since there are 2 motifs that are alternating between activating and repressing effects at different time points, as explained in the next line in the text.

Line 313 - "12 silencers and 60 inhibitors" don't match the numbers in Fig 2D - fixed

Line 342 - should be Fig 3a - fixed

Line 394 - should be Supp Fig 9 - It refers to Supp Fig 8

Line 427 - add percentage (as in line 429) - fixed

Line 451 - clarify what "main categories" you mean - fixed

Line 760 - correct "scrambled" - fixed

Line 827 - correct "RNAD" - fixed

Reviewer #2:

This paper addresses an important question about the basic biology of how enhancers function across different cell types as cell differentiate from iPSCs to NPCs. It leverages lentiMPRA technology, elegantly applied to the differentiating system, which enables the same library of elements to be assessed in different cell types. It is quite clearly written and the authors claim to find several general principles of enhancers from their study of these 591 genomic elements and perturbation of 255 motifs within these, including about 20 double perturbations. They anchor it in the system, time points, and data from their 2019 Cell Stem Cell paper, allowing them good reference back to the cellular context.

Major claims of the paper: 1) There are 4 categories of functional elements. 2) Activity direction is determined by the element itself, but magnitude is determined by the cellular context. 3) Fine tuning of expression levels is mediated by adjacent elements activating and repressing. 4) there sometimes is cooperation between motifs, and both the enhanceosome and billboard model fit the data in some cases.

There are a few spots where I had some statistical concerns that need to be addressed with re-analysis or writing, perhaps one experiment to better query repressive/silencing sequences. Also, the presentation of the data could be clarified in several of the figures, supplemental figures, and especially legends and I have made some suggestions. If the statistical concerns are mitigated by the added analyses and/or experiment, overall I think the paper could be suitable with a substantial revision.

Review:

Substantial concerns:

Line 780: I am concerned that allowing 20% of the barcodes to come from a different sequence might be attenuating some of their results. What happens if you required a 99% of the reads for a barcode to align to a single genomic element? I imagine you would end up with less FRS regions for subsequent analysis, and so a less comprehensive dataset. But, if you do this, are the major conclusions of the paper still the same? If so, then it might be fine to present the 80% data but discuss in methods that the findings are robust to this parameter.

We thank the reviewer for this comment. To address it, we ran our analysis pipeline using a 99% threshold for association. We found that this higher threshold produces transcription rate estimates that are highly consistent with our original values (Pearson's correlation 0.97). We have therefore added the following to the methods section: "To make sure that our results are robust to the association thresholds, we repeated our analysis with a 99% threshold for confident association, which resulted in highly consistent activity estimates (Pearson's correlation 0.97)."

Line 298, and generally their ability to assess silencing sequences. Is it possible the minP is too weak to enable seeing repressive sequences? After all, you need a fair amount of expression present to repress it. Or would very repressive sequences be eliminated from the analysis by filtering (i.e., no RNA counts? – line 790)? Or the ARE's in the construct preventing them from having their effect? Certainly they could try adjusting their filtering to account for the 'no RNA count' filter preventing discovery of repressive elements and see if more are detected. Otherwise, perhaps these points just warrant discussion and a different library design is needed to better understand repression (i.e. no AREs and a higher activity minP).

The reviewer is correct that repressive elements will not be easily detected by our protocol and analysis. While our design identified dampening elements, transcriptionally repressive elements are out-of-scope in this study. We have rephrased the language around the repressive elements in our results to clarify that these are not transcriptional repressors.

Figure 3 (and similar panels on later figures): Why are there no error bars on the individual element perturbations? There are 3 biological replicates as well as multiple barcodes for each? And multiple different ways of perturbing each. There should be a reasonable way to represent some experimental error on these figures.

We thank the reviewer for this comment. All relevant figures have been revised to include error bars, using the multiple replicates.

When motifs really overlap a lot (4d, 6b) how do you interpret perturbations in them? Is it possible to perturb one motif and not the other and did they design to make sure this was the case? How does this change the interpretation relative to the enhanceosome vs billboard models? For example, are the overlapping ones more likely to fall into the enhanceosome model, while more distal ones act more like billboards?

For their discussion section, if neither the enhanceosome model nor the billboard model fit the data, can the authors propose an alternative that does fit all the data?

We thank the reviewer for the opportunity to clarify this section. To address all comments from the reviewers, we completely revised the '**Perturbation of motif pairs identifies different modes of motif interaction**' section by implementing a statistical model to test for pair

interaction and accordingly revised all the relevant parts under the Discussion and Methods sections. We additionally revised Figure 5 (replacing previous Figs. 5-6). This is now revised in the main text and all other relevant parts in the manuscript:

“We considered the FRSs to have independent effects (following the billboard model) if the effects were log-additive: perturbing both sites was equivalent to multiplying the effects of perturbing each site separately. We used MPRAnalyze²² to test this hypothesis for each assayed pair in each perturbation method, by including an interaction term in the model that captures the effect of perturbing both sites while accounting for the effect of perturbing both sites individually (**Methods**). We considered pairs to have significant interaction if the size of the interaction term was larger than 0.5 and the test was statistically significant (BH-corrected $p < 0.05$). We then defined interaction as “consistent” if the pair were labeled the same (either significant or non-significant) in perturbation methods 3 and either 1 or 2, and removed inconsistent pairs from the analysis. We then also removed pairs in which none of the perturbations are functional, by requiring that at least one of the single perturbations pass the filtering scheme we described above. Finally, to make interpreting the results easier, we excluded pairs in which the assayed sites overlap since overlapping sites cannot be conclusively independent. Overall, out of 149 examined pairs, 24 pairs remained, of which 13 were log-additive, consistent with a billboard model of cooperation, and 11 had significant non-additive interactions (**Fig. 5b**). While the small number of functional pairs in our results does not allow for extensive or systemic analyses, we do find anecdotal evidence of different cooperation models operating in different regions.”

Lines 211-214, is there a supplemental figure that goes with this data? Statistical tests to support the assertions?

Supplementary Fig. 5 statistically supports these statements - we now make this clearer in the text by pointing to it directly.

Minor concerns:

Figure 6 sounds like they are thinking about testing for main effects of each mutation, and then testing for an interaction term. Are they? It seems that would be the most straightforward way to be confident when there is an interaction. Is their nested model compatible with including such an interaction term? Then they could confidently state when the interaction between the two perturbations is significant.

We thank the reviewer for the suggestion and have revised the analysis of motif pairs accordingly. We now use MPRAnalyze to test whether the effects of the individual perturbations are log-additive or if they have a statistically significant interaction.

I find the section around line 321/ Supplementary figure 8 hard to follow. For example, the terms a and b on line 319 are not defined. Is everything here in log2FC rather than just FC? In addition, I am confused about the discrepancy between 8a-c (which show a linear relationship) and 8d-e, which argue for a nonlinear relationship using the same data? Are they just showing examples at the low end of expression a-c? Or extrapolating to what would happen beyond expression levels that they measure? What is the major conclusion of this section?

We thank the reviewer for the comment and have clarified the section accordingly. The linear relationship we found is between the WT expression and the absolute difference between the WT and perturbed (PERT) expression: $WT - PERT$. When converted to FC values ($WT / PERT$) this linear relationship translates to a function that saturates and approaches a constant for sufficiently high WT levels. We have added the following clarification to the text: *When examining fold-change values, this linear relationship translates to: $FC = PERT / WT = (1-a) - (b / WT)$ for the same constants. This relationship saturates and approaches a constant (1-a) for sufficiently high levels of unperturbed (WT) expression [...] These constants therefore capture the activation dynamics of each element: a determines the saturated value, and b determines the rate of saturation.*

I assume there is a poly A signal in their reporter too downstream of the WPRE?

There is not a typical poly A signal downstream of the WPRE in our lentiviral vector. It was reported that an internal poly A signal can decrease virus titer (Hager et al., 2008. doi.org/10.1089/hum.2007.165). To better clarify this, we added this note to the Methods section.

Supplemental figures should have legends to make them easier to interpret. I was often left guessing what some panels or axis labels meant. Fonts on figures are often quite small. The authors should be applauded for including thorough supplemental tables that others might use for reanalysis in the future. However, the legends for these should tell potential end-users what the different sheets are, and what the different columns are and how to interpret them (e.g., in the case of coding variables, or the second row of some of the spreadsheets that appears to be some kind of numerical code?).

All the supplementary data and legends have been significantly revised for clarity.

It's a little unclear (e.g., around 295) whether each sequence gets compared to its own scrambled control? Or all scrambled controls?

Each sequence gets compared to all scrambled controls. To better clarify this, we have added the following to the text:

“1) *Essential*: activating sites that when perturbed, reduced the expression level to that of the controls (SCRAM) sequences; “

Very minor/grammatical:

I am trying to figure out what a perturbation MPRA is (line 131). Is it mutating the motifs? Or assaying across time? Either way, I agree it is powerful, but testing expression with and without motif disruptions is what MPRA have been used for a long time. I'm not sure I'd emphasize a new name too much for an old thing. This is not a big deal though.

Perturbation MPRA is the use of MPRA to perturb sequences and assess the effect of these perturbations. The reviewer is correct that this approach has been used in previous work. It has been mainly used to perturb motifs associated with a small number of TFs. The strength of this work is the comprehensiveness of the number of TFs for which motifs are perturbed and the temporal aspect of having the same motif perturbed across multiple time points. Additionally, we take advantage of analysis pipelines we recently developed to find statistically significant effects.

Sentence on line 87 might better be broken into 2 sentences.

We thank the reviewer for the suggestion and have changed the text accordingly.

Line 117 makes it sound like they are evaluating the endogenous target gene. They might want to change this to "reporter gene."

We changed the text accordingly.

The four categories they mention in their abstract seem more pre-defined once I read the paper, rather than discovered from the data as the abstract might be read to imply. They should perhaps rephrase this a bit in the abstract.

We thank the reviewer for this comment, however as far as we know these categories were not previously defined, and were the result of categorizing the effects we saw in our results.

Figure 3 and similar panels in 4: The key took a minute to parse, as it includes both the name of the perturbed element, some unique coded identifier, and the interpretation of the effect (Contributing, Essential, ect). The authors might want to think about how to present this in a way that separates these things a bit and makes it easier to understand the data quickly at a glance. Also, the effect of Nanog was so small I almost didn't notice it in 3b. Fonts got to be quite small as well, and making the lines a bit thicker in their line weight would make it easier to distinguish the colors, which I struggled with. Generally in the paper the data presentation could be polished up a bit.

There are two 6d's in the legend.

Fixed.

272: extra space

Fixed.

374: refs formatted strangely.

Fixed.

There is no supplemental primer table?

We thank the reviewer for their comment. We now added a **Supplementary Table 10** for primers.

Reviewer #3:

Kreimer et al have taken the next step in the productive use of MPRA technology by providing a robust high-throughput framework for perturbing potential regulatory sequences, specifically querying transcription factor (TF) binding sites; by instantiating a perturbation element, a GFP reporter, and a set of filters for the assignment of higher-probability functional regulatory elements in the genome.

The authors explore this approach along 7 timepoints in the progression of human ES cells to neural progenitors; a developmental pathway that has been demonstrated to be a highly fertile realm for transcriptional regulation. This important and valuable step is validated in two ways: the binding sites occur on genes known to be involved in early neural differentiation and reporter readouts are altered along the timecourse of development. This allows the authors to provide insights into the “regulatory architecture” of the journey of stem cells to neural progenitors.

Their approach is easy to follow and allowed for efficient filtering to focus on a more limited set of functionally regulatory sequences (FRS). The classification system in which the FRS were categorized is supported by previous studies and their results provide nuance for the current models of gene expression regulation by distal sequences (ie, the “enhancesome” and “billboard” models). Two strengths of this work are that: 1) it provides a framework for investigating the relationship of TFs within the same regulatory region, which will likely help to understand large and highly complex regulatory regions that are perturbed in developmental disorders, and 2) the procedures could have significant uptake in any lab where such studies are being contemplated. They have also developed previously published tools, ie MPRAflow and MPRAalyze, to assist in utilizing this system.

Three issues with the data presented should be addressed. 1). The term “temporal invariance” (line287), is applied to what seems to refer to whether an element is an activator or a repressor at times examined. But the term literally suggests that time does not play a role; that is, expression is invariant over time which is clearly not the case in the examples shown or from the many investigations that show temporal dynamics vs invariance.

We thank the reviewer for this comment and agree that more accurate language is needed. We removed the term from the statement, which now reads: “This suggests that the direction of the

effect (activating or repressing) of an FRS primarily depends on DNA sequence, and less so on the protein milieu or on other epigenetic properties that change during differentiation.”

2). A second point that seems counterintuitive is that the repressive elements are found to be no lower than baseline/control; one would expect a lower-than baseline impact for any inhibitory elements and this causes concern.

We thank the reviewer for this comment, and indeed the repressive elements we found are only repressive in the sense that they reduce the activation level of the enhancer sequence they reside in, but do not reduce the overall transcription levels of the reporter. This has been clarified in the text: “Importantly, these repressive elements do not lower the baseline transcription rate of the target gene, and are not transcriptional repressors. The repressor FRSs we identified reduce the expression to levels comparable to the baseline levels of the control sequences (SCRAM), but not below it. These repressive elements therefore attenuate the activating function of the enhancer region they reside in, but do not repress the target gene expression below baseline levels.”

3). Since the FRS perturbations are randomly integrated in this system, it’s hard to believe that the impact on enhancer activity will occur in the same manner in its native chromatin environment and here functional validation beyond decreased expression would be very helpful (see below).

We agree that the impact of enhancer perturbation on its activity may not occur in the same manner in its native genomic environment. To validate TF motif function in the genomic context, we further analyzed the binding of TFs using ChIP-seq data (Meissner et al.) and the impact of the TF overexpression on nearby gene expression (see below).

A potential strength of the report would have been to take a known and a novel regulatory element to the stage of functional validation. This step would have provided compelling and demonstrative value to this approach. If this had been demonstrated, then this contribution would have taken the lead in the field; both for detailing the higher throughput analysis with perturbations as detailed, but also with the follow through of a functional validation step which is something that only currently stands as a promissory note; while an outright deliverable is what is desired. As this reviewer sees the field, there has been an evolution of the MPRA technology over the last decade-plus that is in need of coupling with a biological context, and this paper does not take this important step.

The types of validation yielding biological context that one would really like to see would be to include a phenotypic analysis on the impact of these perturbations on neuron differentiation. This would include showing that the TF is actually bound to a given FRS with some temporal specificity that points to a perturbed developmental process. This study should provide some type of evaluation of whether the effect that is seen on the reporter gene/transcribed barcode, actually translates to differences in gene target regulation and phenotypic outcomes.

We thank the reviewer for these important comments. We note that most of the suggestions require extensive experimental validations and therefore exceed the scope of this paper. To address some of the reviewer's suggestions, we attempted to conduct ChIP-seq on hESCs and NPCs (72 hours post neural induction) using antibodies for SOX1, IRX3, BARHL1, and POU3F1, along with H3K27ac (positive control) with multiple replicates. However, none of them except H3K27ac worked significantly, probably because of low TF abundance and weak specificity of the antibodies. Therefore, we decided to analyze relevant publicly available ChIP-seq data to address whether temporal activity of the functional regulatory sites (FRSs) are consistent with TF temporal binding. This data was obtained from (Tsankov et al., 2015) and has ChIP-seq peaks of different TFs in hESC-derived neuroectoderm.

Namely, using computational analysis we find the following: for SOX1 where we have enough results to show statistical significance, we examined the overlap of ChIP-seq data from Meissner et al. We observe significant overlap (fisher exact test) of ChIP-seq peaks of OTX2 and SOX2 factors for FRS vs. filtered regions (i.e. the perturbation is not significant) in which we perturbed SOX1 motifs. This indicates that the signal we are observing using perturbation MPRA is consistent with endogenous binding of the key transcription factors that play pivotal roles in ES-to-neural differentiation (Lodato et al. 2013, Zillar et al., 2014).

In our previous paper, Inoue et al. 2019, we collected RNA-seq data after overexpression of the following TFs: BARHL1, IRX3, LHX5, OTX1/2, PAX6. For the FRSs that contain motifs of these factors, we observe that ~85% of their closest genes are differentially expressed (when comparing to hESC) with adjusted p-value < 0.05. This serves as an additional support of the endogenous functionality of motifs of these factors in these regions.

We have added the following text to our revised manuscript to describe this:

“To address whether temporal activity of the functional regulatory sites (FRSs) are consistent with TF temporal binding using the following 3 strategies: First, we used RNA-seq data from (Inoue et al. 2019) to compare the timing of motif importance with the respective TF expression. Testing this correlation did not show conclusive results. We speculate that this is due to the nature of our analysis which is motif-based, and since similar sequence motifs are not independent, it is likely that the annotation of the FRSs suffers from misclassification of the binding factor. Additionally, even if the exact factor was known, it is not established in current literature that the magnitude of TF gene expression is directly correlated with its regulatory effect, so a strong correlation is not necessarily expected. Second, we examined the overlap of ChIP-seq peaks of different TFs in hESC-derived neuroectoderm (Tsankov et al. 2015) with regions where SOX1 motifs were perturbed. We observe significant overlap (fisher exact test FDR<0.05) of ChIP-seq peaks of OTX2 and SOX2 factors for FRS compared to filtered regions. This indicates that the signal we are observing using perturbation MPRA is consistent with endogenous binding of the key transcription factors that play pivotal roles in ES-to-neural differentiation (Lodato et al. 2013)(Zillar et al. 2015). Finally, we utilized the data we collected in our previous work (Inoue et al. 2019) of RNA-seq following overexpression of these TFs: BARHL1, IRX3, LHX5, OTX1/2, PAX6. For the FRSs that contain motifs of these factors, we

observe that ~85% of their closest genes are differentially expressed (compared to hESC; FDR<0.05). This serves as an additional support of the endogenous functionality of motifs of these factors in these regions.“

Thus, what the authors did not do, which would have created a more enthusiastic uptake from this reviewer, was to address questions such as: What role do FRSs play during neuronal differentiation? Which type of FRS may be responsible for driving cell identity or hESC maintenance and proliferation? Are the TFs found in interacting FRS novel or are they known regulators? For what reason would a FRS need “fine tuning” using activators and silencers in the context of neural development?

We agree with the reviewer that this is very intriguing and discuss some of these points in **Supplementary Note 1**. For example, we mention the following:

“When we perturb an essential motif, by definition, the enhancer is no longer functional in any of the time points, suggesting that these motifs are required for transcription, but not necessarily for determining a specific temporal pattern of the transcription. We were intrigued to see if we could find such condition specific binding motifs in our data. To that end, we looked for FRSs with motifs in (i) late response WT regions (WT alpha - cluster 3; **Supplementary Fig. 12a**) that exhibit their highest perturbation effect in the later time points (Log2FC cluster 3; **Supplementary Fig. 12a**) and specifically, show significant perturbation effects in 72hr NPC state but not in 0hr embryonic stem cell (ESC) state ; (ii) early response WT regions (WT alpha - cluster 1; **Supplementary Fig. 12a**) that exhibit the highest perturbation effect in the early time points (Log2FC cluster 1; **Supplementary Fig. 12a**) and specifically, show significant perturbation effects in 0hr but not in 72hr. We find 37 sequences that are candidates for driving NPC state (i) and 7 sequences that are candidates for driving ESC state (**Supplementary Table 9**). Our premise was that if such motifs are condition (time point) specific they should determine the regulatory activity of a genomic sequence to be in ESC or NPC state. We thus look for enrichment (hypergeometric test FDR<0.05) of such motifs in all 591 WT regions, either in ESCs WTs (cluster 1) or NPC WTs (cluster 3), bearing in my that this is an underpowered test, we only find the following motifs enriched in NPC WT regions: RELA_M4497_1.02 - GGGGATTTCCA, RELB_M6448_1.02 - GGGGGATTTCCA, SP8_1 - GCCACGGCCACT and no motifs were found to be enriched in ESC WT.“

It is my thought that a Nature Comms paper should provide some reasonable answers to these queries. Currently, this paper is an elegant piece of work and should easily find its place in a high profile techniques-oriented journal.

Reviewers' Comments:

Reviewer #1:

Remarks to the Author:

The authors have substantially revised the manuscript to address our concerns. Our main concerns were largely addressed, yet there are a few minor concerns (partly arising during revision) that should be addressed before publication.

Point 1: Rationale and filtering is now much better explained and the addition of Supplemental Figure 7 significantly improved the transparency of the process.

However we still have the following minor concerns that we think should be addressed: The authors mention that they compared the signal of H3K27ac, ATAC-seq, MPRA and mRNA from their previous work with the filtered vs chosen sequences and report a p-value but we could not find the data supporting this result or the methods used. Please add this information.

Point 2/Point 3:

The addition of Supplemental Figure 10 and 11 majorly improves the paper and our understanding of the quality and content of the data set addressing both major concerns 2 and 3.

However the following point still needs to be addressed in order to clarify the results:

Sup. Fig S11 shows that $\sim\frac{3}{4}$ of motifs (left) only appear in 5 or fewer enhancers, and $\sim\frac{3}{4}$ of TFs (right) only appear in 10 or fewer enhancers. This means that the "motifs pie chart" of Fig 2D are largely based on motifs with very few functional instances. This is not clear from the manuscript and we think it can mislead the readers. What would it look like if only using motifs that appear in >5 or >10 enhancers? Likely the percentage of mixed effects will substantially increase based on Sup. Fig S11. At least a similar pie chart using a more stringent minimum number of FRS per motif should be shown in a Supplementary Figure.

Point 4/5/6: Have been addressed by the authors.

Other/Stylistic:

Fig 4A: Please label early and late clusters for clarity

Fig 4A: Color legend is still missing for heatmap

Fig 5B: Please increase font size and label top motif pairs to help the reader

Supp Fig 10/11: It is not possible to read the axis and legend – please increase font size and make sure the resolution is not diminished in the PDF.

There are a lot of issues with spaces in the manuscript.

Lengthiness of manuscript and apparent duplication of paragraphs still there (please see our original comments)

Reviewer #2:

Remarks to the Author:

They addressed my major concerns. I believe the data support the conclusions made and make an interesting and suitable contribution.

Minor:

Line 648: extra period. Also, does filtered referred to filtered in or filtered out? Filtered on what? The 4 filters earlier in the paper? Or on Sox1 motifs? Also, why are you all looking at FRS with Sox1 motifs perturbed? Or is this meant to be a proxy for Sox2 motifs? Generally I don't follow the logic of lines 645-650. I think it is just a re-writing issue though, and not central to their claims. Line 651-655: this is really a result, and not a discussion point. Maybe the whole para should be moved up into results? And perhaps deserves a sup figure, and a bit more explanation. Is there consistency in that if Pax6 is overexpressed, 85% of the genes near the FRSs with Pax6 motifs specifically were differentially expressed? Is this a sig proportion? (i.e, what is the null hypotheses: are 85% of all genes perturbed? Then this is not shocking. If it is only 10% of genes, that 85% is a pretty good odds ratio on a chi-test or something like that). Is it true of each TF independently?

Editorial:

Line 114 "is" -> "was"

Line 273: "H3K27ac, ATAC-seq, MPRA and mRNA of both the closest gene and the motif's associated TF were significantly lower (all time points combined, Wilcoxon p-value < 10⁻¹⁰ 274) in filtered vs. chosen sequences," It is a bit unclear if they mean filtered in or filtered out. Perhaps it might be better to state it as "removed vs. retained"?

Sup Fig 9: it took me some time to figure out that you all were looking at the variation in WT alpha and WT-Pert of an element across the 7 timepoints. It would help to make that explicit here and in the results (i.e., around line 346, maybe add "across time").

Fig 2a – blue in legend does not match blue in barplotFig

Fig 2b – the last letter of your figure legends ran away to a lower line.

Based on the Y-axis, sub fig 9e is log scale, correct?

Sup fig 10 is not readable. Maybe figure out a subset of these to show that make your point?

Sup fig 11 is not readable. But closer to visible than 10.

Line 525: I am not sure I would say that observation is a hybrid of the two models.

Reviewer #3:

Remarks to the Author:

The authors received a series of comments from the reviewers and, to my reading, made a strong effort to address almost all of them. The suggestion of strengthening the paper with experimental validation was seen as work for another paper. Instead, validation work was attempted using ChIPseq for some key molecules but these experiments appeared not to work. The authors fell back on some computational approaches which can be seen as acceptable. There is a concern that the myriad of new analyses conducted by the authors were packaged together as a series of Supplementary Figures rather than picking and choosing the most important analyses and incorporating them in the body of the Results. But all-in-all, this revision seems to address the bulk of the criticisms in a manner that makes the paper more suitable for publication.

We thank the editor and reviewers for their incredibly helpful comments. We revised our manuscript, figures, and supplementary material to reflect these additional changes and new results (all changes in these files are in purple font so that can be easily observed). These are also described in our point-by-point response to reviewers below. Reviewer comments are in black text and our responses are in purple text.

REVIEWER COMMENTS

Reviewer #1 (Remarks to the Author):

The authors have substantially revised the manuscript to address our concerns. Our main concerns were largely addressed, yet there are a few minor concerns (partly arising during revision) that should be addressed before publication.

We thank the reviewer for all their useful comments that helped improve our manuscript.

Point 1: Rationale and filtering is now much better explained and the addition of Supplemental Figure 7 significantly improved the transparency of the process.

However we still have the following minor concerns that we think should be addressed: The authors mention that they compared the signal of H3K27ac, ATAC-seq, MPRA and mRNA from their previous work with the filtered vs chosen sequences and report a p-value but we could not find the data supporting this result or the methods used. Please add this information.

We thank the reviewer for this useful comment, we now added to **Supplementary Table 2** three additional tabs, one per perturbation method, showing the filters that each sequence passed along with the signal of H3K27ac, ATAC-seq and MPRA in this region and mRNA of the closest gene, in all time points. We now revised the main text to include reference to this table as well as the Supplementary Table legend.

Point 2/Point 3:

The addition of Supplemental Figure 10 and 11 majorly improves the paper and our understanding of the quality and content of the data set addressing both major concerns 2 and 3.

However the following point still needs to be addressed in order to clarify the results: Sup. Fig S11 shows that $\sim\frac{3}{4}$ of motifs (left) only appear in 5 or fewer enhancers, and $\sim\frac{3}{4}$ of TFs (right) only appear in 10 or fewer enhancers. This means that the “motifs pie chart” of Fig 2D are largely based on motifs with very few functional instances. This is not clear from the manuscript and we think it can mislead the readers. What would it look like if only using motifs that appear in >5 or >10 enhancers? Likely the percentage of mixed effects will substantially increase based on Sup. Fig S11. At least a similar pie chart using a more stringent minimum number of FRS per motif should be shown in a Supplementary Figure.

We thank the reviewer for this comment, we now added in the main text a clarification on the distribution of motif effects when considering a more stringent number of enhancers they appear in. "We note that most of the motifs in our dataset (~75% **Supplementary Fig. 10**) appear in five or less regions. Constraining the analysis to motifs that appear in more than five regions shows that 16 out of 35 such motifs (~45%) are strictly activators and all of them have mixed effects depending on the region."

Point 4/5/6: Have been addressed by the authors.
We thank the reviewer for these useful comments.

Other/Stylistic:

Fig 4A: Please label early and late clusters for clarity
Fixed.

Fig 4A: Color legend is still missing for heatmap
Fixed.

Fig 5B: Please increase font size and label top motif pairs to help the reader
Font size was increased. We attempted to add readable labels but it resulted in a cluttered figure and in our opinion decreased readability of the figure. The full results are available in **Supplementary Table 8**.

Supp Fig 10/11: It is not possible to read the axis and legend – please increase font size and make sure the resolution is not diminished in the PDF.
To address readability of these figures, we now only visualize the two figures as high resolution PDFs and include the full results as tabs in **Supplementary Table 3**.

There are a lot of issues with spaces in the manuscript.
Lengthiness of manuscript and apparent duplication of paragraphs still there (please see our original comments).
Fixed.

Reviewer #2 (Remarks to the Author):

They addressed my major concerns. I believe the data support the conclusions made and make an interesting and suitable contribution.

We thank the reviewer for all their useful comments that helped improve our manuscript.

Minor:

Line 648: extra period. Also, does filtered refer to filtered in or filtered out? Filtered on what? The 4 filters earlier in the paper? Or on Sox1 motifs? Also, why are you all looking at FRS with Sox1 motifs perturbed? Or is this meant to be a proxy for Sox2 motifs? Generally I don't follow the logic of lines 645-650. I think it is just a re-writing issue though, and not central to their claims.

This has now been re-written in the main text for clarification. We are focusing on SOX1 motif perturbed regions for statistical significance as we have many such instances in our data. "Second, we examined the overlap of ChIP-seq peaks of different TFs in hESC-derived neuroectoderm⁷² with regions where SOX1 motifs were perturbed, for sufficient statistical power. We observe significant overlap (fisher exact test FDR<0.05) of ChIP-seq peaks of OTX2 and SOX2 factors for FRSs compared to regions that were filtered out using the 4 filters described previously. This indicates that the signal we are observing using perturbation MPRA is consistent with endogenous binding of the key transcription factors that play pivotal roles in ES-to-neural differentiation^{39,71}."

Line 651-655: this is really a result, and not a discussion point. Maybe the whole para should be moved up into results? And perhaps deserves a sup figure, and a bit more explanation. Is there consistency in that if Pax6 is overexpressed, 85% of the genes near the FRSs with Pax6 motifs specifically were differentially expressed? Is this a sig proportion? (i.e, what is the null hypotheses: are 85% of all genes perturbed? Then this is not shocking. If it is only 10% of genes, that 85% is a pretty good odds ratio on a chi-test or something like that). Is it true of each TF independently?

We agree with the reviewer that this is an interesting result and we used Fisher's exact test per TF examining its statistical significance. We think that considering the small number of FRSs (<10), these results should remain anecdotal in the discussion. We now added the following text to the manuscript: "Comparing the number of differentially expressed genes that are closest to the FRS to the distribution of the total number of differentially expressed genes, for each overexpressed TF, yielded a statistically significant result for PAX6 (Fisher exact test p-value<0.02). However a larger number of tested FRSs will be needed to make more rigorous conclusions."

Editorial:

Line 114 "is" -> "was"

Fixed.

Line 273: "H3K27ac, ATAC-seq, MPRA and mRNA of both the closest gene and the motif's associated TF were significantly lower (all time points combined, Wilcoxon p-value< 10⁻¹⁰ 274) in filtered vs. chosen sequences," It is a bit unclear if they mean filtered in or filtered out. Perhaps it might be better to state it as "removed vs. retained"?
Fixed.

Sup Fig 9: it took me some time to figure out that you all were looking at the variation in WT alpha and WT-Pert of an element across the 7 timepoints. It would help to make that explicit here and in the results (i.e., around line 346, maybe add "across time").

Fixed.

Fig 2a – blue in legend does not match blue in barplotFig

Fixed.

Fig 2b – the last letter of your figure legends ran away to a lower line.

Fixed.

Based on the Y-axis, sub fig 9e is log scale, correct?

Correct, we fixed the label.

Sup fig 10 is not readable. Maybe figure out a subset of these to show that make your point?

We thank the reviewer for the comment and have now replaced Supplementary Figure 10 with a pdf format for higher resolution, and included the original heatmap in table form as a tab in Supplementary Table 3.

Sup fig 11 is not readable. But closer to visible than 10.

We thank the reviewer and have taken a similar approach to Supplemental Figure 10, in creating a high resolution pdf and including the full results in Supplementary Table 3.

Line 525: I am not sure I would say that observation is a hybrid of the two models.

We changed the wording to “combination” of the two models, as opposed to a single “hybrid” model.

Reviewer #3 (Remarks to the Author):

The authors received a series of comments from the reviewers and, to my reading, made a strong effort to address almost all of them. The suggestion of strengthening the paper with experimental validation was seen as work for another paper. Instead, validation work was attempted using ChIPseq for some key molecules but these experiments appeared not to work. The authors fell back on some computational approaches which can be seen as acceptable. There is a concern that the myriad of new analyses conducted by the authors were packaged together as a series of Supplementary Figures rather than picking and choosing the most important analyses and incorporating them in the body of the Results. But all-in-all, this revision seems to address the bulk of the criticisms in a manner that makes the paper more suitable for publication.

We thank the reviewer for all their useful comments that helped improve our manuscript.

Reviewers' Comments:

Reviewer #1:

Remarks to the Author:

The authors have addressed all our comments and we recommend publication.

Reviewer #2:

Remarks to the Author:

All set. Looks good. Congrats.